# MagicStain: High-Fidelity Pathology Image Virtual Staining via Guided Single-Step Diffusion

## Abstract

Virtual staining, which leverages computational methods to generate different stain styles from a pathology image of a chemically stained tissue section, offers a cost-effective alternative to chemical multiple staining. Despite extensive research based on generative adversarial networks (GANs) and diffusion models, achieving high-fidelity, high-quality, and computationally efficient virtual staining remains a significant challenge. While diffusion-based approaches typically produce more photorealistic images than GAN-based counterparts through multi-step sampling, this comes at the cost of high computational overhead and inference latency. This paper proposes MagicStain, a novel single-step diffusion model tailored for generating high-resolution virtual stains. Specifically, we adapt a pretrained single-step diffusion model to enable efficient virtual staining. By introducing pathology priors from a pretrained pathology-specific vision language model and integrating pathology- and structure-consistency losses on both the original images and the hematoxylin (H)-channel, MagicStain achieves high-fidelity and high-quality generations. To address the limitations of single-step diffusion models in high-resolution virtual staining, we further propose a two-stage progressive training strategy that enables high-resolution adaptation with low training cost. Extensive experiments on three virtual staining datasets, each involving translation between different staining dyes/biomarkers, demonstrate the superiority of MagicStain in terms of fidelity, visual quality, and computational efficiency compared to existing methods. Our code and trained models will be released.

## 1 Introduction

Histochemical staining is widely employed to distinguish various tissues by colorizing them, thus facilitating pathological diagnosis (De Cuyper et al., 2020; Iqbal & Iqbal, 2014; Stack et al., 2014). Different types of dyes manifest different colors in stained tissue and yield complementary information. For instance, hematoxylin-eosin (H&E) highlights cellular morphology, while immuno-histochemistry (IHC) reveals protein-specific expression critical for tumor identification and cancer prognosis (Bai et al., 2023). However, current chemical protocols permit only a single stain per tissue section, necessitating the use of additional sections for multiple stainings. This not only increases the consumption of often-limited clinical tissue samples but also incurs substantial time and chemical costs. As a result, performing multiple stains is resource-intensive, laborious, and expensive (Bai et al., 2023).

Virtual staining (Bai et al., 2023; Khan et al., 2023) offers a promising and cost-effective alternative to conventional chemical staining for generating multiple stains. Being essentially an image-to-image (I2I) translation task, it digitally "translates" chemically stained pathology images using computational methods, converting images stained with one dye (the source domain) to new images that resemble those stained with another (the target domain), e.g., H&E to IHC. Ideally, the virtually stained image should be of high quality (e.g., clear, sharp, and exhibiting the same brightness and color style as the target domain) and high fidelity (accurately representing the authentic content and pathology of the source image).

H&E Input     VIMs     MagicStain (ours)     Ground truth

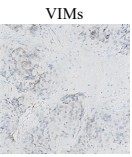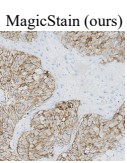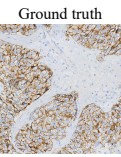

Figure 1: Existing single-step pathology image diffusion model VIMs (Dubey et al., 2024; Parmar et al., 2024) fails to correctly transfer the pathology of source images to the target domain: compared to the ground truth, although the overall distribution and structure of the blue dots (representing nuclei) in the generated IHC image appear hardly realistic, the image lacks the brownish components indicating cancerous regions due to a lack of domain expertise. Our prior-driven MagicStain synthesizes the correct pathology while achieving high structural accuracy.

Researchers have explored generative adversarial networks (GANs; Goodfellow et al., 2014) and diffusion models (Ho et al., 2020) for virtual staining. GAN-based methods utilize image style transfer baselines (e.g., pix2pix (Isola et al., 2017), CycleGAN (Zhu et al., 2017)) to translate the staining style of source-domain images to that of the target domains, while preserving cellular morphology, tissue structure, and pathological semantics. They often incorporate additional modules or loss functions to enforce structure or pathology consistency explicitly, leading to improved fidelity (Peng et al., 2024). Diffusion-based methods (Kataria et al., 2024), built on image-conditional generation baselines such as ControlNet (Zhang et al., 2023), may synthesize images of higher quality via multi-step sampling but suffer from substantial inference latency and computational costs.

To expedite inference and improve the practical applicability of diffusion models, various methods distill slow, many-step teacher models into faster, few-step students (Luo et al., 2023; Xu et al., 2024; Yin et al., 2024). Notably, SD-Turbo (Sauer et al., 2024) distills the Stable Diffusion (SD; Rombach et al., 2022), a multi-step diffusion model, into a one-step model via adversarial training for text-to-image generation. However, directly adding standard diffusion model adapters like ControlNet (Zhang et al., 2023) to the one-step SD-Turbo for I2I translation would cause structure distortion due to the conflict between the noise map and conditioning image (Parmar et al., 2024). Pix2pix-Turbo (Parmar et al., 2024) builds upon SD-Turbo by adopting its architecture and pretrained weights. Unlike SD-Turbo, pix2pix-Turbo enables structure-preserving I2I translation by directly feeding (latents of) source images to the denoising network without an extra adapter, supporting tasks such as day-to-night and foggy-to-clear translation.

Motivated by the superb generation quality and recent advancements in speedy inference of diffusion models, this work aims to achieve one-step, high-resolution virtual staining of pathology images using diffusion models, which has been rarely explored before. However, directly applying pix2pix-Turbo for this purpose, as done in VIMs (Dubey et al., 2024), fails to correctly transfer the pathology of source images to the target domain. Compared to the ground truth (Fig. 1), although the overall structure of the blue dots (representing nuclei) in the IHC images generated by VIMs appears hardly realistic, these images lack the brownish components that indicate cancerous regions (see more results in the Experiments section). We attribute this to pix2pix-Turbo's limited domain expertise in pathology, including pathological semantics and cellular morphology. On the contrary, GAN-based methods widely adopt specific designs to explicitly enforce pathology and structure correctness (e.g., Peng et al., 2024). In addition, VIMs has not explored performance generalization to higher resolutions than 512×512 pixels, which are beneficial and desirable for large-scale images like whole slide imaging (WSI).

This work presents MagicStain, an approach to high-fidelity (in terms of correct structure and pathology) and efficient single-step, high-resolution virtual staining of pathology images. Similar to pix2pix-Turbo, MagicStain adapts pretrained SD-Turbo as its baseline architecture but proposes innovative solutions to address the issues above. To tackle the first issue of lacking 1) domain knowledge and 2) explicit pathology and structure constraints, we design a pathology-prior-aided pipeline. Specifically, we extract pathology priors from source images using PathCLIP (Radford et al., 2021; Sun et al., 2024b), a pretrained vision-language model (VLM) specialized in pathology that serves as a "pathology expert", and inject the priors into the denoising network. Meanwhile, we apply supervision on the similarity between the PathCLIP-extracted pathology priors of the generated and ground-truth target-domain images. In addition, based on the medical prior that the hematoxylin

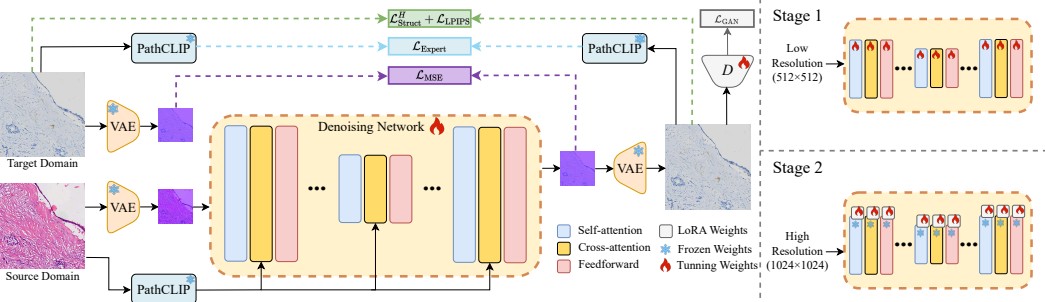

Figure 2: Left: *Framework of the proposed MagicStain*. MagicStain adapts SD-Turbo, a pretrained one-step diffusion model, for single-step pathology image virtual staining. Additionally, it incorporates PathCLIP, a specialized VLM that offers expertise in pathology images. The color-enhanced images serve as schematic illustrations of latent-space representations (the output of the VAE encoder and the input of the VAE decoder) Right: *Progressive training for high-resolution adaptation*. Stage 1 trains all parameters using images of 512×512 pixels. Stage 2 updates only LoRA weights using images of 1024×1024 pixels.

(H)-channel highlights nuclei in H&E and IHC images, we enhance perceptual and reconstruction losses on the H-channel to enforce precise generation of nuclear morphology.

To address the second issue of high-resolution generalization, we propose a progressive training strategy for high-resolution adaptation. We first train the model on low-resolution images for rapid convergence, then fine-tune it on high-resolution images using low-rank adaptation (LoRA; Hu et al., 2022). Our experiments demonstrate that this strategy enables fast generalization from low to high resolution, and achieves better performance than directly training a high-resolution model from scratch or applying a low-resolution model to high-resolution generation.

In summary, our contributions are as follows:

- We introduce MagicStain, a simple yet effective single-step diffusion model that enables high-fidelity and high-resolution virtual staining of pathology images.
- We seamlessly incorporate domain-specific modules and loss functions driven by medical priors, ensuring accurate translation of cellular structures and pathology.
- We propose a progressive training strategy for high-resolution adaptation. To our knowledge, MagicStain is the first single-step diffusion model that synthesizes virtual stains at the resolution of 1024×1024 pixels.
- Last but not least, we conduct extensive experiments on three virtual staining datasets to validate MagicStain against state-of-the-art methods. We also assess a downstream task and perform ablation studies on MagicStain.

## 2 RELATED WORK

**GAN-based Virtual Staining.** Early methods predominantly adopted GANs (Isola et al., 2017; Zhu et al., 2017) for virtual staining of pathology images. (Wang et al., 2024a; Li et al., 2023b; Liu et al., 2022; Klöckner et al., 2025) focused on transferring H&E to IHC images. (Asaf et al., 2024; Hu et al., 2024; Khan et al., 2023; Rivenson et al., 2019) demonstrated the generation of H&E images from formalin-fixed, paraffin-embedded (FFPE) tissue. (Xu et al., 2025; Ma et al., 2023) achieved conversions across other types of pathological images. Most of these GAN-based methods incorporated certain supervision informed by pathology priors—such as classification labels (Jia et al., 2025; Li et al., 2024; Liu et al., 2021; Xu et al., 2019), optical density maps (Chen et al., 2024a), or diaminobenzidine channel supervision (Wei et al., 2024; Peng et al., 2024)—to boost accurate transfer of pathology. Others leveraged loss functions concerning structural consistency, including H-channel similarity (Wei et al., 2024), structure priors from third-party tools (Wang et al., 2024a; Peng et al., 2024; Hu et al., 2024), embeddings of DINO-based histopathology pretrained models (Yang et al., 2025) and auxiliary structural supervision (Wang et al., 2024b; Sloboda et al.,

2023; Dubey et al., 2023; Pati et al., 2024), to enforce consistency between the source and target images in terms of cellular morphology, tissue structures, stromal textures, and spatial relationships. Despite notable progress, GAN-based methods still fall short in generating high-quality, sharp, and realistic pathology images. In this work, we apply the emerging diffusion models to the challenging task of virtual staining.

**Diffusion-based Virtual Staining.** Recent advances in diffusion models (Rombach et al., 2022; Zhang et al., 2023) have demonstrated superior quality in controllable image synthesis to conventional GANs, and have also spurred applications to virtual staining (Ho et al., 2025; Xiong et al., 2025; Zheng et al., 2025; Liu et al., 2025). Großkopf et al. (2025) incorporated third-party models to provide pathology and structure priors. He et al. (2024b) employed an asymmetric attention mechanism to focus on pathological regions and utilized frequency-domain information to preserve structural details. Kataria et al. (2024) generated IHC images and cell segmentation masks in parallel using dual diffusion models to impose morphological supervision. However, all these methods require many-step sampling, resulting in high inference latency and computational overhead. In addition, due to the repetitive denoising process of typical diffusion models (Song et al., 2020), it is difficult to impose direct pathology and structure constraints on the final output, which has been commonly adopted and proven effective in GAN-based approaches. In contrast, our MagicStain addresses both issues using a single-step diffusion model.

To expedite inference for diffusion models, various approaches proposed distilling slow, many-step teacher models into faster, few-step students, including consistency model training (Luo et al., 2023; Song et al., 2023), adversarial learning (Sauer et al., 2024; Xu et al., 2024), variational score distillation (Yin et al., 2024), rectified flow (Liu et al., 2023), and their combinations (Sauer et al., 2024). Some also leveraged model quantization (Li et al., 2023c;d; He et al., 2024a) or pruning (Han et al., 2016) for acceleration. For pathology image virtual staining, VIMs (Dubey et al., 2024) adopted pix2pix-Turbo (Parmar et al., 2024), a single-step I2I translation method, and demonstrated pilot proof-of-concept results. However, VIMs directly applied the pix2pix-Turbo architecture without incorporating any domain knowledge of pathology. Furthermore, it only demonstrated virtual staining at the resolution of 512×512 pixels. By contrast, we introduce a pretrained specialist VLM to provide pathology expertise in MagicStain, and propose a two-stage strategy for effective and efficient training of the single-step diffusion model to generate images of 1024×1024 pixels.

A concurrent work, HARBOR (Chen et al., 2025), also incorporated a pathology VLM for virtual staining and achieved encouraging results for pathology-aware generation. However, its inference incurred substantial computational overhead, requiring 5–10 minutes for an image of 256×256 pixels. In contrast, MagicStain takes less than a second to synthesize an image of 1024×1024 pixels.

## 3 METHOD

Our target is to translate images from a source domain $\mathcal{X} \subset \mathbb{R}^{H \times W \times 3}$ (e.g., H&E) to some desired target domain $\mathcal{Y} \subset \mathbb{R}^{H \times W \times 3}$ (e.g., IHC), where $H$ and $W$ are the height and width of the images, by learning a translation function $\hat{y} = f_G(x) : \mathcal{X} \to \mathcal{Y}$ on a paired dataset $\mathcal{D} = \{(x, y) | x \in \mathcal{X}, y \in \mathcal{Y}\}$. The generated image $\hat{y}$ should be as close as possible to the real target-domain image $y$ in terms of not only color, brightness, and contrast, but also tissue structure and pathology status.

The framework of our method, MagicStain, is shown in Fig. 2. It adapts a pretrained text-to-image one-step diffusion model, SD-Turbo (Sauer et al., 2024), including the denoising network and the variational autoencoder (VAE) (Kingma & Welling, 2022) encoder and decoder. As introduced earlier, adding extra image-conditioning branches, such as ControlNet (Zhang et al., 2023), for one-step I2I translation that requires detailed structure preservation would cause conflicts between the input noise and image condition, where the noise influences the output structure. Therefore, we follow pix2pix-Turbo (Parmar et al., 2024) to directly feed the latent representation of the source image encoded by the VAE encoder to the denoising network, without the noise map or additional branches in conventional many-step diffusion models. The latent representation is forwarded through the denoising network—only once, whose output is decoded by the VAE decoder to produce a re-stained pathology image. In addition, adversarial learning is incorporated to ensure the distribution of generated images is close to that of the target domain.

For effective and efficient single-step virtual staining of pathology images, MagicStain proposes three innovative designs. First, it incorporates PathCLIP, a pretrained VLM specialized in pathology, as an "expert" assistant. On the one hand, PathCLIP encodes source images into pathology prior embeddings to guide the I2I translation via multi-head attention. On the other hand, PathCLIP extracts embeddings from the generated and real target-domain images, with a loss enforcing their similarity. Second, based on the medical prior that the H-channel reflects nuclei morphology in H&E and IHC images, we impose extra structural and semantic consistency on the H-channel to enhance structure and pathology fidelity. Third, we adopt a progressive training strategy for high-resolution adaptation, ensuring high-resolution performance while maintaining low training cost and avoiding generalization degradation. Our empirical results validate the efficacy of MagicStain's designs.

## 3.1 INCORPORATE PATHOLOGY EXPERT PRIOR VIA SPECIALIST VLM.

Although some existing I2I translation methods have introduced various pathology priors for virtual staining, they still lack in-depth and comprehensive expertise in pathology. Recently, VLMs trained on large quantities of pathology image-text pairs have demonstrated strong capabilities in extracting histopathological features critical to clinically relevant downstream tasks (Sun et al., 2024a;b). Inspired by the effectiveness of these VLMs, we adopt PathCLIP (Sun et al., 2024b), a pathology VLM trained on 207K high-quality pathology image-caption pairs via contrastive learning, to provide expert-level pathology prior in our MagicStain framework at both the input and output ends.

**At the input end**, PathCLIP serves as an expert-level assistant, providing extensive prior knowledge about the source image to support clinically correct virtual staining—akin to a seasoned pathologist. Specifically, we utilize the image encoder of PathCLIP to encode the source images into pathology-aware embeddings. The embeddings are used in place of the conditioning textual embeddings (i.e., the controlling prompt in the original SD architecture), serving as the keys and values for multi-head cross-attention in the denoising network. Thus, implicit priors concerning pathology statuses are effectively extracted from the source images and injected into the virtual staining framework.

**At the output end**, we propose enhancing pathological fidelity aided by PathCLIP. Concretely, we apply PathCLIP's image encoder to both the generated image $\hat{y}$ and the ground-truth target-domain image $y$. Then, we compute the cosine similarity between the resulting image embeddings and penalize low similarity with the loss

$$\mathcal{L}_{\text{Expert}} = 1 - f_P(\hat{y}) \cdot f_P(y)/\|f_P(\hat{y})\| \, \|f_P(y)\|, \tag{1}$$

where $f_P$ denotes PathCLIP's image encoder and $\|\cdot\|$ denotes L2 norm.

## 3.2 ENHANCE STRUCTURAL AND SEMANTIC CONSISTENCY WITH MEDICAL PRIOR.

SD-Turbo (Sauer et al., 2024) and pix2pix-Turbo (Parmar et al., 2024) have applied pixel-wise difference losses for effective training of structure-preserving few-step diffusion models. Pix2pix-Turbo additionally incorporates the learned perceptual image patch similarity (LPIPS) loss (Zhang et al., 2018) to enhance perceptual fidelity. Following the successful practice, we impose a mean squared error (MSE) loss in the latent space and an LPIPS loss in the pixel space:

$$\mathcal{L}_{\text{Struct}} = \omega_M \cdot \mathcal{L}_{\text{MSE}}(\hat{z}, z) + \omega_L \cdot \mathcal{L}_{\text{LPIPS}}(\hat{y}, y), \tag{2}$$

where $\hat{z}$ is the latent predicted by the denoising network, $z = f_V(y)$ is $y$'s latent representation encoded by the VAE encoder $f_V$, and $\omega_L$ and $\omega_M$ denote the weights for the two losses. In addition, based on the *medical prior that the H-channel highlights nuclei in H&E and IHC images*, we extract the H-channel from the generated and ground truth target-domain images, and impose extra consistency losses to enhance structure and pathology fidelity:

$$\mathcal{L}_{\text{Struct}}^H = \omega_M^H \cdot \mathcal{L}_{\text{MSE}}^H(\hat{y}^H, y^H) + \omega_L^H \cdot \mathcal{L}_{\text{LPIPS}}^H(\hat{y}^H, y^H), \tag{3}$$

where $\hat{y}^H$ and $y^H$ denote the H-channel of $\hat{y}$ and $y$, respectively, $\omega_M^H$ and $\omega_L^H$ are corresponding weights. This targeted supervision, driven by medical prior, explicitly enhances the nuclear structures in the generated images.

## 3.3 ADVERSARIAL TRAINING.

Adversarial learning has demonstrated prominent effectiveness in reducing the sampling steps of diffusion models (Sauer et al., 2024; Parmar et al., 2024; Xu et al., 2024). To adapt the text-to-

image generator SD-Turbo (Sauer et al., 2024) for single-step pathology I2I translation, we also incorporate a vision-aided GAN (Kumari et al., 2022) with a frozen DINOv2 (Oquab et al., 2023) backbone, where the discriminator $f_D$ aims to maximize

$$\mathcal{L}_{\text{GAN}}^D = \log f_D(y) + \log[1 - f_D(\hat{y})]. \tag{4}$$

Meanwhile, the single-step generator $f_G$ (comprising the VAE and the denoising network) is trained to minimize

$$\mathcal{L}_{\text{GAN}} = -\log f_D(\hat{y}) = -\log f_D\big(f_G(x)\big). \tag{5}$$

Thus, the generator learns to generate pathology images matching the distribution of the target domain, i.e., indistinguishable from the real target-domain images by the discriminator. Collectively, the total loss for end-to-end training of the generator is:

$$\mathcal{L} = \omega_E \cdot \mathcal{L}_{\text{Expert}} + \mathcal{L}_{\text{Struct}} + \mathcal{L}_{\text{Struct}}^H + \omega_G \cdot \mathcal{L}_{\text{GAN}}, \tag{6}$$

where $\omega_E$ and $\omega_G$ are constant weights.

### 3.4 ADAPT FOR HIGH-RESOLUTION SINGLE-STEP STAINING.

Considering the ultra-large-scale nature of WSIs, high-resolution generation is desirable for reducing the number of patches required to represent a complete WSI. However, diffusion methods, while capable of generating high-quality images, may suffer from prohibitive computational costs and convergence difficulties when scaled to high-resolution training. Meanwhile, although models trained at the resolution of $512 \times 512$ pixels can be directly used for inferring images of $1024 \times 1024$ pixels, the performance may be suboptimal due to the lack of adaptation for higher-resolution representations.

We observe that models trained with images of $512 \times 512$ pixels already acquire core virtual staining abilities, i.e., correctly transferring pathology and structure in general. Yet, their generalization to higher resolutions is limited due to incompatible latent spaces. Hence, such models only need lightweight adaptation for high resolutions. Accordingly, we propose a progressive strategy for training a 1024-resolution virtual staining model efficiently. In the first stage, we train the model on images of $512 \times 512$ pixels, where the lower computational demand and task complexity allow for rapid convergence. In this stage, the denoising network is fully fine-tuned to enable effective learning of cellular structures and pathological semantics, which are fundamental to the virtual staining task.[1]. Therefore, we keep the pretrained VAE frozen to reduce training cost. Then, in the second stage, we introduce LoRA (with rank=8 following (Parmar et al., 2024)) into the denoising network and fine-tune only the LoRA weights using images of $1024 \times 1024$ pixels. The rationale is to quickly adapt the model for high-resolution with only a small number of parameters, but not interfering with the core virtual staining capabilities acquired in the first stage. Empirical results in Table 3 demonstrate that with the same training epochs, our strategy consistently achieves superior virtual staining quality.

## 4 EXPERIMENTS

**Datasets and Evaluation Metrics.** We evaluate MagicStain on three datasets for comprehensive performance assessment across various virtual staining tasks. RegH2I (Peng et al., 2024) contains 2,592 pairs of registered images for H&E-to-IHC staining, where the IHC images are labeled with the HER2 biomarker (H&E2IHC-HER2). (Hu et al., 2024) provided 5,098 aligned image pairs for FFPE-to-H&E staining (FFPE2H&E). H&E2IHC-CK, a dataset we collected, comprises 2,557 aligned image pairs for H&E-to-IHC staining, utilizing high molecular weight cytokeratin (CK) as the IHC biomarker. We follow the official train/validation/test splits for FFPE2H&E and H&E2IHC-HER2. For H&E2IHC-CK, we randomly allocate 241 pairs as the test set and use the remaining 2,316 pairs for training. All evaluations are conducted on cropped patches of $1024 \times 1024$ pixels in accordance with (Hu et al., 2024; Peng et al., 2024).

Following existing practice (Peng et al., 2024; Hu et al., 2024), we report the structural similarity index measure (SSIM) and peak signal-to-noise ratio (PSNR) to reflect the quality of virtually

---

[1]Our empirical results showed that fine-tuning the VAE led to similar performance (see Appendix C.4 for details)

stained images. Typically, higher SSIM values indicate a better structural match between the virtually stained and ground truth images. However, due to intrinsic structural discrepancies in paired histopathology images stained with different dyes, exact pixel-wise structural correspondence is generally infeasible, even after registration (more explanation in Appendix A). This phenomenon is more pronounced in the FFPE2H&E dataset. Therefore, we also report the Fréchet inception distance (FID; Heusel et al., 2017) and kernel inception distance (KID; Bińkowski et al., 2018), which are better aligned with human perceptual quality, for a more comprehensive evaluation. Notably, lower FID and KID values indicate a smaller disparity between the distributions of generated results and the target domains.

**Implementation.** All experiments are conducted using Ubuntu 20.04, Python 3.10.16, and PyTorch 2.4.0 (Paszke et al., 2019) on an NVIDIA A800 GPU with 80 GB memory. Following pix2pix-Turbo (Parmar et al., 2024), we adopt pretrained SD-Turbo (Sauer et al., 2024) as the backbone of our MagicStain. We use the AdamW optimizer (Loshchilov & Hutter, 2017) with a learning rate of $5 \times 10^{-6}$. We train the model with batch sizes of two and one for the resolutions of $512 \times 512$ and $1024 \times 1024$ pixels, respectively. We first train the model for 15 epochs using images of $512 \times 512$ pixels and then fine-tune it for five additional epochs with images of $1024 \times 1024$ pixels, where the former is obtained by evenly dividing the latter into four patches. The loss weights are empirically set as follows: $\omega_M = 1$; $\omega_L = 4$; $\omega_L^H = 4$; $\omega_M^H = 0.5$; $\omega_E = 0.2$; and $\omega_G = 0.5$. Our code and trained models will be released.

**Comparison with State-of-the-Art (SOTA).** We compare MagicStain with classical GAN-based image translation methods: CycleGAN (Zhu et al., 2017), pix2pix (Isola et al., 2017), and pix2pixHD (Wang et al., 2018); SOTA diffusion models for image translation: ControlNet (Zhang et al., 2023) and pix2pix-Turbo (Parmar et al., 2024); and specialized SOTA approaches to pathology image virtual staining,[2] including both GAN-based: (Peng et al., 2024), (Hu et al., 2024), and AI-FFPE (Ozyoruk et al., 2021), and diffusion-based: StainFuser (Jewsbury et al., 2024). As ControlNet and StainFuser are multi-step Diffusion models, we adopt their default inference steps in our experiments (50 and 20, respectively). Comparisons with (Peng et al., 2024) and (Hu et al., 2024) are exclusively on H&E2IHC (including -HER2 and -CK) and FFPE2H&E, respectively, as each is tailored for its corresponding task.

*Quantitative analysis.* As Table 1 shows, MagicStain achieves the best performance across all metrics on the H&E2IHC-HER2 and H&E2IHC-CK datasets, indicating superior perceptual and structural quality of its virtually stained images. In Table 2, MagicStain shows slightly lower PSNR and SSIM than the top-performing methods but remains highly competitive (among the top three) on FFPE2H&E. We speculate this is due to the relatively weaker pixel- and structure-level correspondence between paired images in FFPE2H&E than the other datasets, which affects structural evaluation. Nevertheless, MagicStain achieves the best scores for the perception-oriented metrics—FID and KID, demonstrating its superior capability in pathology-correct virtual staining. We also benchmark the inference time of the compared diffusion models for generating one sample: ControlNet 18.69 seconds, StainFuser 8.92 seconds, pix2pix-Turbo 0.85 seconds, and our MagicStain 0.84 seconds. This comparison highlights the advantage of single-step diffusion models in terms of inference efficiency.

*Qualitative analysis.* Additionally, we present qualitative results for all three datasets, providing insightful analysis. Fig. 3 shows the results on H&E2IHC-HER2 (the results on H&E2IHC-CK and FFPE2H&E are provided in Appendix C.1). As we can see, MagicStain outperforms other methods in structural accuracy by exhibiting more precise cellular structures and clearer cell morphology. This is notably demonstrated in Case 1, where blue dots denote cell nuclei in the IHC-stained image: the distributions and sizes of the nuclei in MagicStain are most consistent with those in the ground truth. Furthermore, MagicStain displays more faithful pathological characteristics. This is evident in Case 2, where brown regions in the IHC images reflect cancerous lesions, with darker brown indicating more pronounced lesions and more advanced disease stages. MagicStain closely matches the ground truth in both lesion location and cancer severity. In contrast, compared methods either underestimate (e.g., pix2pix-Turbo) or overestimate (e.g., ControlNet) the presence/severity of cancer, which is worsened by imprecise lesion locations. The unsatisfactory pathology statuses

---

[2]In our context, the virtual staining model VIMs (Dubey et al., 2024) embodies a straightforward application of pix2pix-Turbo. Therefore, we only list pix2pix-Turbo in our comparison.

|  | FID↓ | KID↓ | PSNR↑ | SSIM↑ | FID↓ | KID↓ | PSNR↑ | SSIM↑ |
|---|---|---|---|---|---|---|---|---|
| Method | | | H&E2IHC-HER2 | | | | H&E2IHC-CK | |
| Pix2pix | 47.77 | 0.0237 | $18.04^*{\pm}3.693$ | $0.401^*{\pm}0.126$ | 51.49 | 0.0026 | $19.12^*{\pm}2.695$ | $0.489^*{\pm}0.103$ |
| Pix2pixHD | 41.77 | 0.0123 | $18.06^*{\pm}3.733$ | $0.386^*{\pm}0.128$ | 45.44 | 0.0014 | $19.10^*{\pm}2.921$ | $0.482^*{\pm}0.111$ |
| (Peng et al., 2024) | 33.92 | 0.0058 | $18.02^*{\pm}3.706$ | $0.385^*{\pm}0.125$ | 50.01 | 0.0028 | $18.94^*{\pm}2.668$ | $0.476^*{\pm}0.103$ |
| CycleGAN | 40.91 | 0.0062 | $17.01^*{\pm}3.524$ | $0.365^*{\pm}0.119$ | 190.3 | 0.1427 | $15.73^*{\pm}3.427$ | $0.443^*{\pm}0.122$ |
| AI-FFPE | 37.15 | 0.0047 | $17.46^*{\pm}3.764$ | $0.385^*{\pm}0.122$ | 57.86 | 0.0027 | $17.32^*{\pm}3.297$ | $0.440^*{\pm}0.120$ |
| ControlNet | 209.7 | 0.2560 | $7.368^*{\pm}0.591$ | $0.127^*{\pm}0.021$ | 110.0 | 0.0498 | $12.20^*{\pm}1.052$ | $0.313^*{\pm}0.076$ |
| StainFuser | 104.5 | 0.0791 | $17.30^*{\pm}3.949$ | $0.401^*{\pm}0.148$ | 119.9 | 0.0653 | $13.49^*{\pm}3.598$ | $0.324^*{\pm}0.156$ |
| pix2pix-Turbo | 225.2 | 0.2909 | $17.91^*{\pm}3.316$ | $0.344^*{\pm}0.114$ | 56.36 | 0.0050 | $18.40^*{\pm}2.997$ | $0.489^*{\pm}0.114$ |
| MagicStain (ours) | **33.51** | **0.0033** | **18.58±3.668** | **0.411±0.125** | **44.70** | **0.0011** | **19.49±3.017** | **0.505±0.110** |

Table 1: Evaluation of various methods on the H&E2IHC-HER2 and H&E2IHC-CK datasets. *: $p < 0.05$ by the Wilcoxon signed-rank test for pairwise comparison with our method.

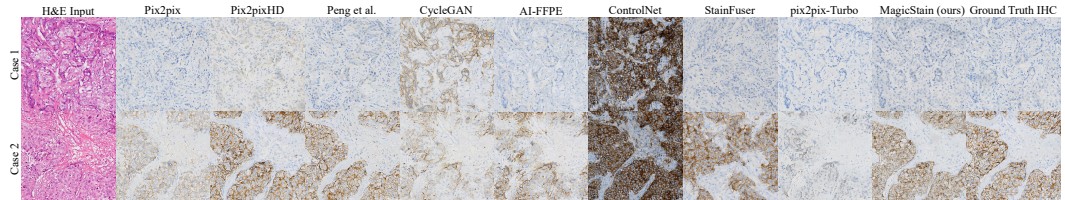

Figure 3: Example virtual staining results on H&E2IHC-HER2 (best viewed when zoomed in). In the IHC images, brown regions indicate cancerous lesions, with darker brown coloration indicating more pronounced malignancy and a more advanced disease stage.

of the images generated by ControlNet and pix2pix-Turbo are in accordance with their high FIDs in Table 1. In conclusion, the promising structure and pathology fidelity of MagicStain imply the effectiveness of incorporating domain VLM and medical prior.

**External Validation.** To further evaluate the generalizability of our method, we conduct further validation on the external test set from (Peng et al., 2024), which includes 285 H&E-IHC image pairs stained with SP3 (D'Alfonso et al., 2013) or CB11 (Purdie et al., 2010) antibodies (H&E2IHC-Ext). Notably, we directly apply the models trained on the H&E2IHC-HER2 dataset in the main experiments without further fine-tuning. As shown in Table 4, MagicStain again achieves the best results in FID, KID, and SSIM, on the independent patient cohorts, while remaining comparable with other methods in PSNR. These results align well with the main experiments in Table 1, further validating the strong generalizability of MagicStain in virtual staining tasks.

**Evaluation on Downstream Task.** To examine the practical usability of virtually stained IHC images in downstream clinical tasks, we conduct a membrane staining intensity classification experiment using the H&E2IHC-HER2 and H&E2IHC-Ext datasets, following the HER2 scoring guideline (Ahn et al., 2020) and the same protocol as in (Peng et al., 2024). This task involves categorizing four levels of membrane staining intensity. For each virtual staining method, we use 600 patches of virtual IHC images re-stained from the H&E images in the test set of H&E2IHC-HER2, to train a ResNet-50 classifier (He et al., 2016). The classifier is subsequently evaluated on the real IHC images of H&E2IHC-Ext to compute accuracy and F1 score. For reference, we have also trained a classifier using H&E images and another one using real IHC images in the test set of H&E2IHC-HER2. As shown in Table 5, the classifier trained on IHC images generated by MagicStain outperforms those trained on images by all compared methods. It even surpasses the upper-bound classifier trained on real IHC images, whose performance might be affected by the domain gap between the H&E2IHC -HER2 and -Ext datasets. We also evaluate these classifiers on an in-domain dataset (please refer to Section C.11 for details). Combining these results, the classifier trained on IHC images generated by MagicStain achieves performance comparable to that obtained using real IHC images. Meanwhile, the classifier trained on H&E images performs poorly due to its inability to convey membrane staining intensity, as expected. These findings underscore the clinical relevance and potential of accurate virtual staining from H&E to IHC, particularly in facilitating reliable assessment of HER2 expression.

| Method | FID↓ | KID↓ | PSNR↑ | SSIM↑ |
|---|---|---|---|---|
| Pix2pix | 17.88 | 0.0008 | 18.34*±2.009 | 0.536*±0.113 |
| Pix2pixHD | 15.26 | 0.0008 | 19.08±2.046 | 0.586±0.106 |
| (Hu et al., 2024) | 31.56 | 0.0101 | **19.86±2.082** | **0.644±0.098** |
| CycleGAN | 35.36 | 0.0087 | 14.47*±2.045 | 0.363*±0.152 |
| AI-FFPE | 25.59 | 0.0023 | 13.92*±1.971 | 0.355*±0.148 |
| ControlNet | 156.0 | 0.1039 | 11.50*±0.882 | 0.187*±0.069 |
| StainFuser | 64.98 | 0.0301 | 12.46*±1.851 | 0.263*±0.157 |
| pix2pix-Turbo | 22.33 | 0.0022 | 16.93*±1.697 | 0.505*±0.113 |
| MagicStain (ours) | **13.98** | **0.0001** | 18.52±2.082 | 0.544±0.112 |

Table 2: Evaluation of various methods on the FFPE2H&E dataset. *: $p < 0.05$ by Wilcoxon signed-rank test for pairwise comparison with our method.

| Method | FID↓ | KID↓ | PSNR↑ | SSIM↑ |
|---|---|---|---|---|
| w/o prior | 34.24 | 0.0044 | 18.38*±3.651 | 0.401*±0.127 |
| w/o $\mathcal{L}_{\text{Expert}}$ | 35.33 | 0.0042 | 18.53*±3.679 | 0.407*±0.128 |
| w/o $\mathcal{L}_{\text{MSE}}$ | 34.40 | 0.0046 | 18.38*±3.700 | 0.405*±0.125 |
| w/o $\mathcal{L}_{\text{LPIPS}}$ | 34.98 | 0.0049 | 18.20*±3.580 | 0.387*±0.124 |
| w/o $\mathcal{L}_{\text{MSE}}^{H}$ | 35.84 | 0.0070 | 18.43*±3.612 | 0.409*±0.124 |
| w/o $\mathcal{L}_{\text{LPIPS}}^{H}$ | 35.38 | 0.0051 | 18.42*±3.642 | 0.400*±0.126 |
| w/ $\mathcal{L}_{\text{MSE}}^{I}$ | 34.40 | 0.0042 | 18.51*±3.580 | 0.409*±0.123 |
| w/o 1024 | 43.88 | 0.0135 | 18.40*±3.885 | 0.409*±0.133 |
| w/o 512 | 35.01 | 0.0053 | 18.41*±3.587 | 0.408*±0.128 |
| w/o LoRA | 38.15 | 0.0069 | 18.50*±3.707 | **0.417±0.131** |
| MagicStain (ours) | **33.51** | **0.0033** | **18.58±3.668** | 0.411±0.125 |

Table 3: Ablation study on the H&E2IHC-HER2 dataset. *: $p < 0.05$ by Wilcoxon signed-rank test for pairwise comparison with our method.

| Method | FID↓ | KID↓ | PSNR↑ | SSIM↑ |
|---|---|---|---|---|
| Pix2pix | 147.0 | 0.1065 | 16.63±3.442 | 0.325*±0.126 |
| Pix2pixHD | 104.9 | 0.0597 | 16.36±3.390 | 0.341±0.128 |
| (Peng et al., 2024) | 104.7 | 0.0796 | 15.70*±2.234 | 0.345±0.130 |
| CycleGAN | 149.6 | 0.1168 | **16.91±3.484** | 0.341±0.117 |
| AI-FFPE | 92.89 | 0.0365 | 16.66±3.235 | 0.323*±0.122 |
| ControlNet | 235.9 | 0.2741 | 7.245*±0.491 | 0.107*±0.021 |
| StainFuser | 156.5 | 0.0932 | 16.13±3.336 | 0.343±0.143 |
| pix2pix-Turbo | 289.2 | 0.3261 | 16.61±3.413 | 0.328*±0.116 |
| MagicStain (ours) | **92.24** | **0.0271** | 16.17±3.229 | **0.348±0.135** |

Table 4: Evaluation on an external validation dataset (H&E2IHC-Ext). For all methods, the models trained on H&E2IHC-HER2 are directly used here for inference without any tuning. *: $p < 0.05$ by Wilcoxon signed-rank test for pairwise comparison with our method.

| Method | ACC↑ | F1↑ |
|---|---|---|
| H&E | 0.4246 | 0.3205 |
| Real IHC | 0.7228 | 0.7298 |
| Pix2pix | 0.5930 | 0.5253 |
| Pix2pixHD | 0.6386 | 0.5430 |
| (Peng et al., 2024) | 0.7263 | 0.6795 |
| CycleGAN | 0.5263 | 0.4714 |
| AI-FFPE | 0.6912 | 0.6251 |
| ControlNet | 0.3965 | 0.3202 |
| StainFuser | 0.6448 | 0.5807 |
| pix2pix-Turbo | 0.3509 | 0.2070 |
| MagicStain (ours) | **0.7404** | **0.7459** |

Table 5: Evaluation of a downstream classification task using the H&E2IHC-HER2 and H&E2IHC-Ext datasets.

**Ablation Studies.** This section ablates various designs of MagicStain on the HER2-HE2IHC dataset to validate their rationality, and collectively presents the results in Table 3. Concerning "pathology expert prior", we propose injecting pathology priors extracted by the specialist VLM into the denoising network via multi-head cross-attention. To evaluate the efficacy of this prior, we ablate it by replacing it with an empty embedding (denoted by "w/o prior"). We also ablate the VLM-based expert similarity loss $\mathcal{L}_{\text{Expert}}$ (Eqn. (1)) by setting $\omega_E = 0$ (w/o $\mathcal{L}_{\text{Expert}}$). Results show that removing either the pathology prior or the associated loss $\mathcal{L}_{\text{Expert}}$ degrades FID and KID, confirming the value of the expertise provided by the specialist VLM.

To investigate the contributions of the structure and pathology consistency losses, we ablate the MSE and LPIPS losses imposed on the entire images ("w/o $\mathcal{L}_{\text{MSE}}$" and "w/o $\mathcal{L}_{\text{LPIPS}}$", respectively), as well as the corresponding losses on the H-channel ("w/o $\mathcal{L}_{\text{MSE}}^{H}$" and "w/o $\mathcal{L}_{\text{LPIPS}}^{H}$", respectively), by setting their weights to 0. It turns out that removing any of them consistently degrades all evaluation metrics. In particular, removing the MSE and LPIPS losses on the H-channel, which highlights nuclei in the target-domain images, results in more pronounced drops in pathology-relevant perceptual metrics (FID and KID). Meanwhile, removing the corresponding losses on the entire image leads to more noticeable degradation in structural metrics. Therefore, both types of structural and pathological supervision are necessary for the optimal performance of MagicStain.

For $\mathcal{L}_{\text{MSE}}$, we further experiment with computing MSE in the image space ("w/ $\mathcal{L}_{\text{MSE}}^{I}$"), instead of the latent space. We find that latent-space supervision yields slightly better results, likely because feature values in the latent space span a broader range, thus providing stronger gradient signals.

For adaptation to high-resolution single-step staining, we propose a two-stage progressive training strategy: first, fine-tuning all parameters with images of $512\times512$ pixels for 15 epochs, followed by LoRA fine-tuning with images of $1024\times1024$ pixels for an additional five epochs. We compare this strategy to two alternatives: training only with images of $512\times512$ or $1024\times1024$ pixels (w/o 1024 or 512), for 20 epochs, with all parameters updated. Results show that training at a single resolution consistently degrades performance: training only at 512 resolution leads to a generalization issue when applied to the higher resolution of 1024, while training only at 1024 resolution is difficult to converge. These results validate our design for high-resolution single-step staining.

Lastly, to determine the necessity of LoRA, we also ablate it by updating all parameters in the second (high-resolution adaptation) training stage (w/o LoRA). As we can see, while PSNR and SSIM remain comparable, both FID and KID degrade substantially. This suggests that, compared with LoRA, fine-tuning all parameters during high-resolution adaptation may cause the model to lose its fundamental virtual staining capability acquired in the first stage.

## 5 CONCLUSION

This paper presented MagicStain, a high-fidelity pathology image virtual staining model via single-step diffusion guided by a specialized VLM. Thorough experiments validated its promising performance and effective design. We plan to extend MagicStain for more virtual staining tasks.

## ETHICS STATEMENT

All experiments in this study were conducted using pathological slides collected from real clinical scenarios, with proper authorization obtained from the relevant hospitals. No patient-identifiable information is included in the dataset. This study does not involve clinical trials, interventions with human subjects, or any new dataset releases beyond those authorized. There are no potential conflicts of interest or sponsorships associated with this work. The paper does not raise issues related to privacy, security, discrimination, bias, fairness, legal compliance, or research integrity.

## REPRODUCIBILITY STATEMENT

We are committed to ensuring the reproducibility of our work. Detailed descriptions of our approach are provided in the Section 4. All code, trained models, and necessary scripts will be released to the public in the near future to facilitate further research.

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

APPENDIX

## A    MORE EXPLANATION OF NON-PERFECT PAIRED DATA

In virtual staining tasks, paired images are typically obtained from consecutive tissue sections or chemical re-staining of the same tissue section. The former results in two different tissue sections, and the latter process inevitably deforms the tissue section between two stains. Therefore, it is generally infeasible to achieve perfect registration. While the alignment does not achieve perfect pixel-wise matching, the majority of pixels are well-aligned, making them suitable for learning structural correspondences. Prior to our approach, numerous methods have effectively trained virtual staining models using the same datasets employed in this study (Hu et al., 2024; Peng et al., 2024).

## B    MORE INFORMATION ON DATASETS

The H&E2IHC-HER2 and FFPE2H&E datasets were collected from breast cancer patient cohorts, whereas the H&E2IHC-CK dataset was collected from a prostate cancer cohort. Therefore, our MagicStain has been evaluated on datasets of two organs/cancer types. The consistently superior performance of MagicStain compared to other methods across organs and cancer types highlights its robustness and generalization ability.

It is worth noting that we did not use the public datasets such as BCI (Liu et al., 2022) and MIST (Li et al., 2023a) due to their poor registration quality between paired images, as discussed in HEMIT (Bian et al., 2024). Both BCI and MIST suffer from severe misalignment and noticeable variations in tissue morphology between source and target domains. These issues may potentially compromise the learning of correct mapping between images of different stains, resulting in unreliable evaluations of virtual staining results.

## C    ADDITIONAL EXPERIMENTS

### C.1    ADDITIONAL QUALITATIVE RESULTS.

We present additional qualitative results across all three datasets, H&E2IHC-HER2, H&E2IHC-CK, and FFPE2H&E, under the same experimental settings as Tables 1 and 2, and Figure 3 in the main text. Figure 4 shows that MagicStain exhibits clearer tissue structures, more precise cellular morphology, and more faithful pathological characteristics than other methods.

### C.2    CHOICE OF PATHOLOGY PRETRAINED MODEL.

For the "expert" pathology pretrained model that provides pathology expertise in MagicStain, we have adopted PathCLIP (Sun et al., 2024b)—a vision-language model (VLM) pretrained via contrastive learning between textual and visual embeddings—in the main text. Here, we further experiment with an additional pathology VLM, PathGenCLIP (Sun et al., 2024a), as well as DINO-based self-supervised histopathology models, including UNI2-h (Chen et al., 2024b) and Virchow2 (Zimmermann et al., 2024). We replace PathCLIP with PathGenCLIP, UNI, or Virchow on the H&E2IHC-HER2 dataset for both I2I guidance and pathology-accuracy supervision.

Table 6 presents the performance on the H&E2IHC-HER2 dataset. MagicStain with PathGenCLIP achieves results comparable to PathCLIP and outperforms all other methods in Table 1, indicating that both VLMs encode strong pathology priors for virtual staining. In contrast, UNI2-h and Virchow2 perform worse than PathCLIP and PathGenCLIP. We attribute this to the semantic priors of the pathology VLMs (PathCLIP and PathGenCLIP) that are aligned with human textual descriptions, which provide more suitable pathology-relevant signals than DINO-based models (UNI2-h and Virchow2) and thus offer more appropriate cues for understanding pathology in the source domain. Despite that, using UNI or Virchow still yields performance competitive with leading methods in Table 1, which suggests that MagicStain can effectively incorporate other histopathology pretrained models to achieve consistently strong performance.

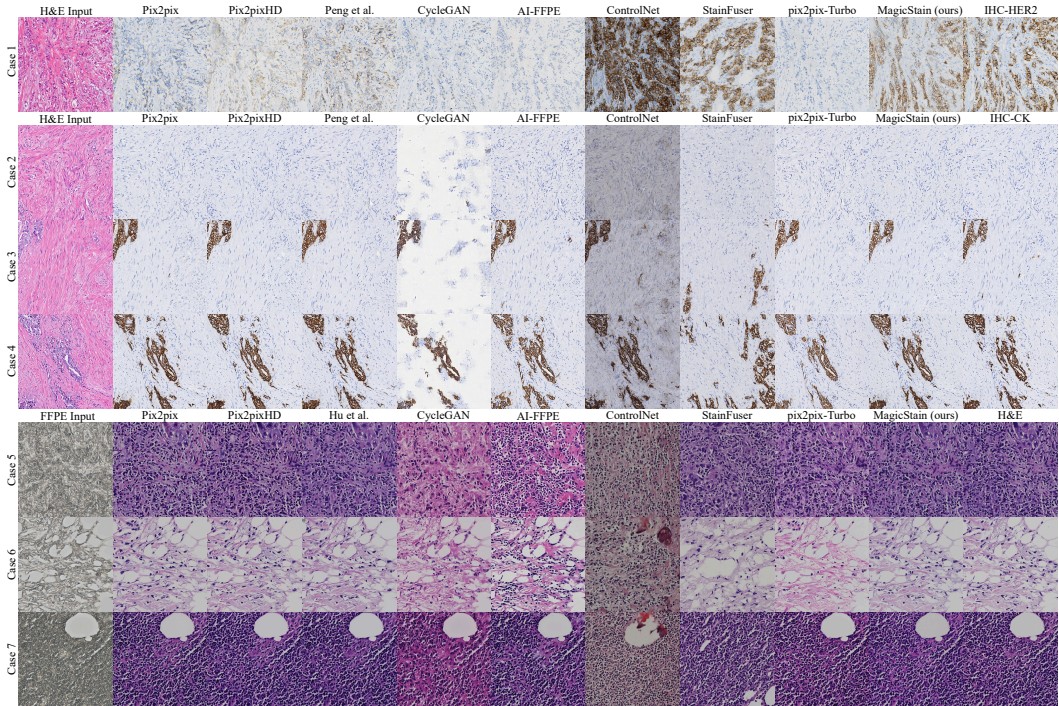

Figure 4: Example virtual staining results by various methods on three tasks: Case 1: H&E2IHC-HER2; Cases 2–4: H&E2IHC-CK; and Cases 5–7: FFPE2H&E (best viewed when zoomed in). In IHC-HER2 and IHC-CK images, brown regions indicate cancerous lesions. In particular, in the IHC-HER2 image, darker brown coloration signifies higher malignancy and a more advanced disease stage. In H&E images, dark purple dots represent cell nuclei, while light purple or reddish regions correspond to the cytoplasm. Enlarged nuclei may indicate malignant cells.

| VLM | FID↓ | KID↓ | PSNR↑ | SSIM↑ |
|---|---|---|---|---|
| UNI2-h | 33.84 | 0.0064 | 18.50±3.721 | 0.408±0.126 |
| Virchow2 | 33.94 | 0.0050 | 18.32±3.624 | 0.402±0.127 |
| PathGenCLIP | 33.52 | 0.0034 | 18.65±3.750 | 0.410±0.131 |
| PathCLIP (ours) | 33.51 | 0.0033 | 18.58±3.668 | 0.411±0.125 |

Table 6: Performance comparison of using various pathology pretrained models as the expert on the H&E2IHC-HER2 dataset.

| First stage | FID↓ | KID↓ | PSNR↑ | SSIM↑ |
|---|---|---|---|---|
| LoRA | 77.36 | 0.0472 | $17.67^*$±3.253 | $0.395^*$±0.117 |
| Full (ours) | **33.51** | **0.0033** | **18.58**±3.668 | **0.411**±0.125 |

Table 7: Comparison between LoRA and full-parameter fine-tuning (ours) in the first training stage (using images of 512×512 pixels) on the HE2IHC-HER2 Dataset. *: $p < 0.05$ by Wilcoxon signed-rank test for pairwise comparison.

## C.3 SHOULD BOTH STAGES ADOPT LOW-RANK ADAPTATION (LoRA)?

In the main text, we have proposed progressive low-to-high resolution adaptation, where the first stage updates all parameters of the denoising network using images of 512×512 pixels and the second stage implements LoRA using images of 1024×1024 pixels. This strategy has demonstrated a major contribution in MagicStain, as revealed by our ablation study in Table 3. Here, we explore the question: can LoRA be used in the first stage, too, for more computationally efficient training? This means that throughout the low- and high-resolution training, only the LoRA parameters are updated. Experimental results are shown in Table 7. Compared with full-parameter fine-tuning, using LoRA in the first stage results in substantial performance drops in all evaluation metrics. These results suggest that, to endow the baseline text-to-image one-step diffusion model pretrained on natural images with the core virtual staining capabilities, extensive parameter update rather than LoRA is needed.

## C.4 Influence of Fine-Tuning Variational Autoencoder (VAE).

MagicStain uses the frozen, pretrained VAE from Stable Diffusion (SD; Rombach et al., 2022) for pathology image encoding and reconstruction. This section empirically studies the impact of fine-tuning the VAE (with LoRA, similar to (Parmar et al., 2024) ) on performance, using the H&E2IHC-HER2 task. As shown in Table 8, fine-tuning the VAE leads to marginally better FID but worse KID, PSNR, and SSIM. These results suggest that despite the potential domain gap between natural images and pathology images, the pretrained VAE may already suffice for pathology image encoding and reconstruction in MagicStain, and the gap has a minimal impact on performance in this case. We attribute this to the SD VAE's strong encoding/decoding ability acquired from large-scale, diverse training, which is likely to contain pathology images or similar ones. To reduce training cost, we therefore opt not to further train the VAE.

| VAE status | FID↓ | KID↓ | PSNR↑ | SSIM↑ |
|---|---|---|---|---|
| Fine-tuning | **32.33** | 0.0048 | $18.45\pm3.752$ | $0.410\pm0.132$ |
| Frozen (ours) | 33.51 | **0.0033** | **18.58**$\pm3.668$ | **0.411**$\pm0.125$ |

Table 8: Ablation study results on the H&E2IHC-HER2 dataset evaluating the impact of fine-tuning the VAE.

## C.5 Ablation Studies on Loss Weights.

Our total training loss in Eqn. (6) is dependent on a few weights, so is the model performance. Therefore, we have determined their values to optimize model performance, based on empirical evidence on the H&E2IHC-HER2 dataset. We group the weights into three groups: (i) $\omega_E$ and $\omega_G$ for the VLM-based expert similarity and GAN losses, (ii) $\omega_M$ and $\omega_L$ for the original structural and semantic consistency losses in Eqn. (2), and (iii) $\omega_M^H$ and $\omega_L^H$ for the H-channel enhanced structural and semantic consistency losses in Eqn. (3), and vary either weight in a group while fixing all other ones. The rough candidate ranges for the weights were determined based on the corresponding or similar weight values in (Parmar et al., 2024). Tables 10, 11, and 9 present the experimental results. As we can see, while PSNR and SSIM do not show significant variations with different weight values, FID and KID, which are better aligned with human perceptual quality, exhibit noticeable variations. Based on the general principle of choosing the weight value that consistently ranks among the top two across all evaluated metrics, we set $\omega_E = 0.2$; $\omega_G = 0.5$; $\omega_M = 1$; $\omega_L = 4$; $\omega_M^H = 0.5$; $\omega_L^H = 4$, which represent a balance of various losses.

| Weight | FID↓ | KID↓ | PSNR↑ | SSIM↑ |
|---|---|---|---|---|
| $\omega_M^H = 0$ | 35.84 | 0.0070 | $18.43^*\pm3.612$ | $0.409^*\pm0.124$ |
| $\omega_M^H = 0.25$ | 35.34 | 0.0066 | $18.36^*\pm3.570$ | $0.409^*\pm0.125$ |
| $\omega_M^H = 0.5$ | **33.51** | **0.0033** | **18.58**$\pm3.668$ | **0.411**$\pm0.125$ |
| $\omega_M^H = 1$ | 35.09 | 0.0046 | $18.38^*\pm3.582$ | $0.408^*\pm0.125$ |
| $\omega_M^H = 2$ | 34.77 | 0.0063 | $18.47^*\pm3.554$ | $0.409^*\pm0.124$ |
| $\omega_L^H = 0$ | 35.38 | 0.0051 | $18.42^*\pm3.642$ | $0.400^*\pm0.126$ |
| $\omega_L^H = 1$ | 35.02 | 0.0045 | $18.26^*\pm3.611$ | $0.403^*\pm0.127$ |
| $\omega_L^H = 2$ | 34.70 | 0.0051 | $18.31^*\pm3.637$ | $0.405^*\pm0.127$ |
| $\omega_L^H = 4$ | **33.51** | **0.0033** | **18.58**$\pm3.668$ | 0.411$\pm0.125$ |
| $\omega_L^H = 8$ | 35.70 | 0.0043 | $18.49^*\pm3.717$ | **0.415**$\pm0.127$ |

Table 9: The impact of varying $\omega_M^H$ and $\omega_L^H$ on performance on the H&E2IHC-HER2 dataset. *: $p < 0.05$ by Wilcoxon signed-rank test for pairwise comparison with $\omega_M^H = 0.5$ (top) and $\omega_L^H = 4$ (bottom).

## C.6 Additional Experiments on Public Datasets.

We present additional quantitative results on widely used public datasets, including BCI (Liu et al., 2022) and four MIST (Li et al., 2023a) datasets (HER2, ER, Ki67, PR), under the same experimental settings as Tables 1 in the main text.

As shown in Tables 13, 14, and 12 MagicStain achieves the best performance across most of metrics (only little lower PSNR compared with Pix2pix on MIST-PR and SSIM compared with StainFuser on MIST-Ki67) on the five public datasets, demonstrating superior perceptual and structural quality

| Weight | FID↓ | KID↓ | PSNR↑ | SSIM↑ |
|---|---|---|---|---|
| $\omega_E = 0$ | 35.33 | 0.0042 | 18.53*±3.679 | 0.407*±0.128 |
| $\omega_E = 0.1$ | 34.35 | 0.0073 | 18.39*±3.588 | 0.409*±0.125 |
| $\omega_E = 0.2$ | **33.51** | **0.0033** | **18.58**±3.668 | **0.411**±0.125 |
| $\omega_E = 0.4$ | 34.25 | 0.0072 | 18.34*±3.568 | 0.407*±0.125 |
| $\omega_E = 0.8$ | 34.84 | 0.0059 | 18.37*±3.581 | 0.406*±0.125 |
| $\omega_G = 0.25$ | 35.43 | 0.0061 | 18.57*±3.670 | **0.412**±0.129 |
| $\omega_G = 0.5$ | **33.51** | **0.0033** | **18.58**±3.668 | 0.411±0.125 |
| $\omega_G = 1.0$ | 36.31 | 0.0071 | 18.17*±3.701 | 0.403*±0.127 |
| $\omega_G = 2.0$ | 36.40 | 0.0056 | 18.06*±3.678 | 0.391*±0.125 |

Table 10: The impact of varying $\omega_E$ and $\omega_G$ on performance on the H&E2IHC-HER2 dataset. *: $p < 0.05$ by Wilcoxon signed-rank test for pairwise comparison with $\omega_E = 0.2$ (top) and $\omega_G = 0.5$ (bottom).

| Weight | FID↓ | KID↓ | PSNR↑ | SSIM↑ |
|---|---|---|---|---|
| $\omega_M = 0$ | 34.40 | 0.0046 | 18.38*±3.700 | 0.405*±0.125 |
| $\omega_M = 0.25$ | **33.24** | 0.0055 | 18.44*±3.775 | 0.407*±0.125 |
| $\omega_M = 0.5$ | 34.02 | 0.0053 | 18.40*±3.736 | 0.412±0.129 |
| $\omega_M = 1$ | 33.51 | **0.0033** | 18.58±3.668 | 0.411±0.125 |
| $\omega_M = 2$ | 34.34 | 0.0070 | **18.58**±3.652 | **0.416**±0.128 |
| $\omega_L = 0$ | 34.98 | 0.0049 | 18.20*±3.580 | 0.387*±0.124 |
| $\omega_L = 1$ | 35.55 | 0.0047 | 18.27*±3.631 | 0.403*±0.125 |
| $\omega_L = 2$ | 34.80 | 0.0041 | 18.34*±3.715 | 0.404*±0.129 |
| $\omega_L = 4$ | **33.51** | **0.0033** | **18.58**±3.668 | 0.411±0.125 |
| $\omega_L = 8$ | 33.85 | 0.0047 | 18.48*±3.637 | **0.413**±0.127 |

Table 11: The impact of varying $\omega_M$ and $\omega_L$ on performance on the H&E2IHC-HER2 dataset. *: $p < 0.05$ by Wilcoxon signed-rank test for pairwise comparison with $\omega_M = 1$ (top) and $\omega_L = 4$ (bottom).

| Method | FID↓ | KID↓ | PSNR↑ | SSIM↑ |
|---|---|---|---|---|
| Pix2pix | 91.06 | 0.0580 | 21.28±4.773 | 0.544±0.172 |
| Pix2pixHD | 88.81 | 0.0583 | 19.47±5.657 | 0.532±0.172 |
| (Peng et al., 2024) | 63.18 | 0.0258 | 19.96±5.263 | 0.517±0.169 |
| CycleGAN | 61.09 | 0.0304 | 16.46±5.858 | 0.511±0.166 |
| AI-FFPE | 51.31 | 0.0226 | 15.09±4.837 | 0.491±0.147 |
| ControlNet | 236.9 | 0.2381 | 9.100±2.864 | 0.248±0.065 |
| StainFuser | 80.49 | 0.0519 | 13.54±4.243 | 0.519±0.174 |
| pix2pix-Turbo | 303.1 | 0.4019 | 13.64±4.494 | 0.443±0.160 |
| MagicStain (ours) | **30.24** | **0.0037** | **22.38**±4.252 | **0.550**±0.183 |

Table 12: Evaluation of various methods on the BCI datasets.

in its virtually stained images as well as excellent generalization across datasets. In addition, our code and trained models will be released to facilitate reproducible research and further exploration in this direction by the community.

## C.7 ADDITIONAL COMPARISON WITH EXISTING METHODS.

In this section, we compare our MagicStain with two additional state-of-the-art approaches to virtual staining, Pyramidpix2pix (Liu et al., 2022) and ASP (Li et al., 2023a) on the H&E2IHC-HER2 dataset (Peng et al., 2024). As shown in Tables 15, MagicStain achieves better performance across all metrics than Pyramidpix2pix (Liu et al., 2022) and ASP (Li et al., 2023a).

## C.8 VERIFYING TRAINING CONVERGENCE.

To demonstrate that MagicStain can converge within only 15-20 epochs, we conduct an additional set of experiments by training MagicStain for 200 epochs (150 for stage 1 and 50 for stage 2), following the same ratio as Section 4 in the main text (15 and 5 epochs). As shown in Tables 16, the 200-epoch model yields results comparable to those obtained with 20 epochs, indicating that MagicStain has already converged at 20 epochs. This rapid convergence is attributed to MagicStain's domain-specific modules, medically motivated loss functions, and a progressive training strategy

| | FID↓ | KID↓ | PSNR↑ | SSIM↑ | FID↓ | KID↓ | PSNR↑ | SSIM↑ |
|---|---|---|---|---|---|---|---|---|
| Method | | | MIST-HER2 | | | | MIST-ER | |
| Pix2pix | 75.97 | 0.0427 | 15.02±2.855 | 0.249±0.084 | 70.53 | 0.0382 | 15.071±3.209 | 0.269±0.093 |
| Pix2pixHD | 61.45 | 0.0236 | 14.59±2.967 | 0.243±0.085 | 58.45 | 0.0213 | 14.468±3.192 | 0.263±0.095 |
| (Peng et al., 2024) | 50.95 | 0.0148 | 14.58±2.971 | 0.241±0.087 | 54.68 | 0.0204 | 14.352±3.164 | 0.255±0.093 |
| CycleGAN | 67.78 | 0.0343 | 13.91±2.621 | 0.228±0.077 | 46.66 | 0.0142 | 13.113±2.516 | 0.247±0.093 |
| AI-FFPE | 57.41 | 0.0217 | 13.37±2.433 | 0.225±0.076 | 72.54 | 0.0413 | 14.065±2.738 | 0.254±0.080 |
| ControlNet | 170.2 | 0.1244 | 8.335±1.394 | 0.135±0.024 | 245.6 | 0.2156 | 7.3965±1.209 | 0.124±0.020 |
| StainFuser | 88.51 | 0.0490 | 11.11±1.819 | 0.254±0.092 | 98.33 | 0.0615 | 10.094±1.168 | 0.263±0.099 |
| pix2pix-Turbo | 152.5 | 0.1018 | 14.04±2.634 | 0.214±0.072 | 79.73 | 0.0371 | 13.458±2.406 | 0.209±0.079 |
| MagicStain (ours) | **36.71** | **0.0049** | **15.12**±2.917 | **0.257**±0.093 | **30.52** | **0.0032** | **15.074**±3.287 | **0.275**±0.103 |

Table 13: Evaluation of various methods on the MIST-HER2 and MIST-ER datasets.

| Method | FID↓ | KID↓ | PSNR↑ | SSIM↑ | FID↓ | KID↓ | PSNR↑ | SSIM↑ |
|---|---|---|---|---|---|---|---|---|
| | | | MIST-Ki67 | | | | MIST-PR | |
| Pix2pix | 64.99 | 0.0441 | $15.21\pm_{2.146}$ | $0.290\pm_{0.101}$ | 75.11 | 0.0420 | $\mathbf{15.41}\pm_{3.534}$ | $0.286\pm_{0.105}$ |
| Pix2pixHD | 53.29 | 0.0331 | $14.59\pm_{2.162}$ | $0.284\pm_{0.100}$ | 60.08 | 0.0239 | $14.79\pm_{3.563}$ | $0.276\pm_{0.106}$ |
| (Peng et al., 2024) | 44.95 | 0.0223 | $14.64\pm_{2.184}$ | $0.278\pm_{0.100}$ | 65.80 | 0.0323 | $14.82\pm_{3.520}$ | $0.262\pm_{0.100}$ |
| CycleGAN | 38.77 | 0.0122 | $13.70\pm_{1.908}$ | $0.252\pm_{0.089}$ | 45.69 | 0.0113 | $13.40\pm_{2.919}$ | $0.239\pm_{0.086}$ |
| AI-FFPE | 36.98 | 0.0090 | $14.04\pm_{1.946}$ | $0.275\pm_{0.086}$ | 49.93 | 0.0193 | $13.48\pm_{3.060}$ | $0.256\pm_{0.089}$ |
| ControlNet | 250.3 | 0.2851 | $8.272\pm_{1.324}$ | $0.170\pm_{0.031}$ | 272.9 | 0.2609 | $7.585\pm_{1.162}$ | $0.148\pm_{0.028}$ |
| StainFuser | 66.67 | 0.0408 | $10.97\pm_{1.307}$ | $\mathbf{0.312}\pm_{0.100}$ | 80.15 | 0.0461 | $10.26\pm_{1.530}$ | $0.279\pm_{0.124}$ |
| pix2pix-Turbo | 161.5 | 0.1529 | $13.49\pm_{1.458}$ | $0.232\pm_{0.080}$ | 78.62 | 0.0436 | $14.89\pm_{3.126}$ | $0.259\pm_{0.095}$ |
| MagicStain (ours) | $\mathbf{26.13}$ | $\mathbf{0.0034}$ | $\mathbf{15.25}\pm_{2.199}$ | $0.297\pm_{0.105}$ | $\mathbf{29.94}$ | $\mathbf{0.0030}$ | $15.36\pm_{3.540}$ | $\mathbf{0.290}\pm_{0.113}$ |

Table 14: Evaluation of various methods on the MIST-Ki67 and MIST-PR datasets.

| Method | FID↓ | KID↓ | PSNR↑ | SSIM↑ |
|---|---|---|---|---|
| ASP | 65.51 | 0.0294 | $17.30\pm_{2.966}$ | $0.368\pm_{0.104}$ |
| Pyramidpix2pix | 47.52 | 0.0183 | $18.21\pm_{3.722}$ | $0.389\pm_{0.124}$ |
| MagicStain (ours) | $\mathbf{33.51}$ | $\mathbf{0.0033}$ | $\mathbf{18.58}\pm_{3.668}$ | $\mathbf{0.411}\pm_{0.125}$ |

Table 15: Evaluation of ASP (Li et al., 2023a) and Pyramidpix2pix (Liu et al., 2022) on the H&E2IHC-HER2 dataset.

designed for high-resolution adaptation. These designs enable the model achieves high-fidelity, high-resolution virtual staining and surpassing Pix2pix (Isola et al., 2017) and CycleGAN (Zhu et al., 2017) models that require 200 epochs of training, as demonstrated in Sections 4. However, longer training significantly increases computational and energy costs. Thus, we consider 20 epochs sufficient.

| Epochs | FID↓ | KID↓ | PSNR↑ | SSIM↑ |
|---|---|---|---|---|
| 200 | 31.76 | 0.0060 | $18.55\pm_{3.754}$ | $0.415\pm_{0.128}$ |
| 20 | 33.51 | 0.0033 | $18.58\pm_{3.668}$ | $0.411\pm_{0.125}$ |

Table 16: Comparison between MagicStain models trained for 200 and 20 epochs on the H&E2IHC-HER2 dataset.

## C.9 IMPACT OF DAB-CHANNEL ENHANCEMENT.

An IHC image is composed of two main staining components: a Hematoxylin (H) component highlighting nuclear morphology and a DAB component encoding the spatial distribution of the target biomarker. The brown DAB channel in IHC images highlights the location and intensity of target protein expression (e.g., HER2 on the cell membrane and CK in epithelial cells; darker brown indicates stronger expression), which primarily encodes pathological information. In the main text, MagicStain enhances perceptual and reconstruction losses on the H-channel to enforce precise nuclear morphology generation, and relies on PathCLIP (Sun et al., 2024b) to supervise pathological accuracy when generating H&E or IHC images. As demonstrated in Table 3, these components are effective.

Here, we also explore the impact of enhancing the DAB-channel, which serves a role similar to PathCLIP within MagicStain. We conduct experiments on the H&E2IHC-HER2 dataset by adding perceptual and reconstruction losses on the DAB-channel instead of the H-channel (w DAB w/o H) or by using both (w DAB w H). The results of Table 17 show that jointly enhancing the DAB- and H-channels yields performance similar to not enhancing the DAB-channel (slightly better FID and PSNR, and slightly worse KID and SSIM). We attribute this to MagicStain has incorporated global supervision across the entire image (Hematoxylin (H) channel or DAB channel) through an MSE loss in the latent space and a LPIPS loss in the pixel space, which also ensures the accuracy of the DAB channel. However, removing H-channel enhancement (w DAB w/o H) results in clearly inferior performance, indicating that H-channel enhancement is crucial for MagicStain.

Furthermore, the DAB-channel loss can be used only for generating IHC images and cannot be applied to FFPE-to-H&E staining, limiting the method's applicability across multiple stain types. In contrast, the H-channel, which highlights nuclei in both H&E and IHC images, can be used

for both generation tasks. Therefore, MagicStain enhances the fidelity of pathological and nuclear morphology by imposing H-channel loss and PathCLIP similarity loss for H&E and IHC image generation.

| Method | FID↓ | KID↓ | PSNR↑ | SSIM↑ |
|---|---|---|---|---|
| w DAB w H | 33.18 | 0.0049 | 18.60±3.675 | 0.405±0.126 |
| w DAB w/o H | 35.72 | 0.0086 | 18.31±3.597 | 0.403±0.125 |
| w/o DAB w H (Ours) | 33.51 | 0.0033 | 18.58±3.668 | 0.411±0.125 |

Table 17: Impact of enhancing DAB-channel on the H&E2IHC-HER2 dataset.

## C.10 IMPACT OF PRETRAINED WEIGHT.

MagicStain adapts pretrained SD-Turbo (Sauer et al., 2024) as its backbone, like many diffusion applications which use pretrained weights on generic natural images as initialization for achieving better performance. In this section, we also evaluate the performance of training without pretrained weights—otherwise under the same setting as Table 1 in the main text. As shown in the Table 18, removing pretrained weights leads to worse results compared with using pretrained initialization. Nevertheless, it still achieves highly competitive performance compared with existing SOTA methods in Table 1 (only inferior to (Peng et al., 2024)), owing to our architectural advancements—including explicit incorporation of domain knowledge and high-resolution adaptation.

| Method | FID↓ | KID↓ | PSNR↑ | SSIM↑ |
|---|---|---|---|---|
| w/o pretrain | 37.12 | 0.0081 | 18.28±3.708 | 0.407±0.122 |
| w pretrain (Ours) | 33.51 | 0.0033 | 18.58±3.668 | 0.411±0.125 |

Table 18: Impact of pretrained weight on the H&E2IHC-HER2 dataset.

## C.11 DOWNSTREAM TASK PERFORMANCE ON IN-DOMAIN DATA.

In Section 4, the classifier trained on IHC images generated by MagicStain even surpasses the upperbound classifier trained on real IHC images, whose performance might be affected by the domain gap between the H&E2IHC-HER2 and -Ext datasets. The IHC images in H&E2IHC-HER2 are processed using the 4B5 antibody, while the IHC images in H&E2IHC-Ext are stained with the SP3 or CB11 antibodies. As we train the classifiers on 4B5 IHC images (either real or virtually stained), these images are considered in-domain, whereas the SP3 and CB11 IHC images constitute cross-domain data. To evaluate the in-domain performance, we additionally collect 285 4B5 IHC images (the same number of images as in the H&E2IHC-Ext dataset) with no overlap with the H&E2IHC-HER2 data, forming an independent in-domain test set. We then evaluate the in-domain performance by applying the 4B5-trained classifiers to the in-domain test set. As shown in Table 19, the performance is overall better than that on the cross-domain data, as expected. In addition, the classifier trained on MagicStain-generated 4B5 IHC images achieves performance competitive with that of the classifier trained on real 4B5 IHC images. This indicates that the IHC images generated by MagicStain exhibit accurate HER2 expression, i.e., accurate pathological translation from H&E to IHC images.

| Method | ACC↑ | F1↑ |
|---|---|---|
| Real IHC | 0.8456 | 0.8160 |
| MagicStain (ours) | 0.8421 | 0.8171 |

Table 19: Evaluation of the downstream classification task using in-domain IHC images.

## C.12 ADDITIONAL EVALUATION METRICS.

To enable a more comprehensive pathological evaluation, we additionally employ a pathology-specific encoder—PLIP (Zuo et al., 2024)—to assess the gap between the generated results and the target domains by computing the mean cosine similarity of the feature embeddings extracted by PLIP (Zuo et al., 2024) for each pair on the H&E2IHC-HER2 dataset. We also replace the Inception network used for computing FID in Section 4 with UNI2-h (Chen et al., 2024b) to ensure

that the metric calculation is better aligned with the target domain, following (Bhosale et al., 2025). As shown in Table 20, MagicStain achieves the best PLIP similarity and UNI2-h-based FID scores, demonstrating its superior capability in clinically correct virtual staining and validating its effective design and strong performance under pathology-standard evaluation. These results are consistent with the conclusions of ours experiments of virtual staining and downstream tasks.

| Method | Pix2pix | Pix2pixHD | (Peng et al., 2024) | CycleGAN | AI-FFPE | ControlNet | StainFuser | pix2pix-Turbo | MagicStain (ours) |
|---|---|---|---|---|---|---|---|---|---|
| PLIP↑ | 0.9711* | 0.9649* | 0.9724 | 0.9641* | 0.9574* | 0.8367* | 0.9357* | 0.9572* | **0.9733** |
| FID-UNI2-h↓ | 107.12 | 109.52 | 90.883 | 90.593 | 87.001 | 323.28 | 145.21 | 186.39 | **77.764** |

Table 20: PLIP similarity and UNI2-h-based FID scores of various methods on the H&E2IHC-HER2 dataset. *: $p < 0.05$ by Wilcoxon signed-rank test for pairwise comparison with our method.

### C.13 EVALUATION OF SEGMENTATION METRICS FOR ABLATION STUDIES.

We also use segmentation metrics to more extensively evaluate the impact of encoder choices and the H-channel losses in this section, following the methodology in (Kataria et al., 2025). We deconvolute IHC images into hematoxylin and DAB, a brown chromogen (i.e., IHC-positive), and compare the DAB masks between real and virtual images. Specifically, we compute DICE score, intersection over union (IoU), true positive rate (TPR), and true negative rate (TNR) on the H&E2IHC-HER2 dataset. Higher values of these segmentation metrics reflect better alignment of DAB-mask between real and virtual images, indicating improved pathological accuracy.

As shown Table 21, removing the pathology prior ("w/o prior") or its associated loss ("w/o $\mathcal{L}_{\text{Expert}}$"), which enforce pathology-related consistency—degrades the pathological accuracy of the generated results. Replacing PathCLIP with UNI2-h or Virchow2 for both I2I guidance and pathology-accuracy supervision leads to slight drops across all metrics, demonstrating that the pretrained pathology VLM (PathCLIP) provides more appropriate pathology-relevant priors than DINO-based histopathology pretrained models (UNI2-h and Virchow2). Removing losses imposed on the entire image—MSE and LPIPS ("w/o $\mathcal{L}_{\text{MSE}}$" and "w/o $\mathcal{L}_{\text{LPIPS}}$") also harms performance.

Removing the corresponding losses on the H-channel ("w/o $\mathcal{L}_{\text{MSE}}^{H}$" and "w/o $\mathcal{L}_{\text{LPIPS}}^{H}$") yields results comparable to those obtained when including them ("w/o DAB w H (Ours)"), indicating that enhancement of the H-channel do not lead to inaccurate DAB-channel. However, as shown in Table 3, removing the MSE and LPIPS losses on the H-channel, which highlights nuclei in the target-domain images, results in pronounced drops in perceptual metrics (FID and KID) and structural metrics (PSNR and SSIM). Meanwhile, explicitly emphasizing the DAB channel ("w DAB w H" and "w DAB w/o H") does not yield additional gains, as PathCLIP already provides sufficient guidance. Therefore, both H-channel losses and the supervision combined with the PathCLIP prior are essential for achieving the optimal performance of MagicStain.

| Method | TPR↑ | TNR↑ | DICE↑ | IoU↑ |
|---|---|---|---|---|
| w/o prior | 0.3579* | 0.8462* | 0.3271* | 0.2523* |
| w/o $\mathcal{L}_{\text{Expert}}$ | 0.3587* | 0.8444* | 0.3248* | 0.2498* |
| UNI2-h | 0.3663* | 0.8523* | 0.3306* | 0.2533* |
| Virchow2 | 0.3667* | 0.8546* | 0.3325* | 0.2569* |
| w/o $\mathcal{L}_{\text{MSE}}$ | 0.3593* | 0.8446* | 0.3245* | 0.2499* |
| w/o $\mathcal{L}_{\text{LPIPS}}$ | 0.3492* | 0.8425* | 0.3283* | 0.2492* |
| w/o $\mathcal{L}_{\text{MSE}}^{H}$ | 0.3733 | 0.8525* | 0.3320* | 0.2556* |
| w/o $\mathcal{L}_{\text{LPIPS}}^{H}$ | 0.3722 | 0.8598 | 0.3334 | 0.2558 |
| w DAB w H | 0.3775 | 0.8546* | 0.3347 | 0.2584 |
| w DAB w/o H | 0.3743 | 0.8550* | 0.3369 | 0.2582 |
| w/o DAB w H (Ours) | 0.3717 | 0.8567 | 0.3346 | 0.2590 |

Table 21: Segmentation Metrics of Ablation Studies on the H&E2IHC-HER2 dataset. *: $p < 0.05$ by Wilcoxon signed-rank test for pairwise comparison with our method.

### C.14 DOWNSTREAM TASKS QUALITATIVE RESULTS.

In this section, we provide qualitative examples of the downstream task for Section 4, including the images of H&E2IHC-Ext to be classified in Figure 5, and the classification results of various methods in Table 22. In HER2 scoring, "0" denotes no cancerous lesion, while "1+", "2+", and

Case 1  Case 2  Case 3  Case 4

Figure 5: H&E or IHC images to be classified from H&E2IHC-Ext, as shown in Section C.14. HER2 scores: 0 indicates no cancerous lesion, while 1+, 2+, and 3+ represent increasing severity of cancerous lesions. Cases 1 to 4 correspond to HER2 scores of 0, 1+, 2+, and 3+, respectively.

| Method | H&E | Pix2pix | Pix2pixHD | Peng et al. (2024) | CycleGAN | AI-FFPE | ControlNet | StainFuser | pix2pix-Turbo | MagicStain | GT |
|--------|-----|---------|-----------|---------------------|----------|---------|------------|------------|---------------|------------|-----|
| Case 1 | **3+** | **1+** | 0 | 0 | **2+** | 0 | **3+** | 0 | 0 | 0 | 0 |
| Case 2 | **3+** | **1+** | 1+ | 1+ | **0** | **0** | **0** | **0** | **0** | 1+ | 1+ |
| Case 3 | **3+** | **3+** | **3+** | **3+** | **0** | 2+ | **3+** | 2+ | **0** | 2+ | 2+ |
| Case 4 | 3+ | 3+ | 3+ | 3+ | **2+** | 3+ | **0** | 3+ | **1+** | 3+ | 3+ |

Table 22: Classification results for each case in Figure 5 for the different classifiers. HER2 scores: 0 indicates no cancerous lesion, while 1+, 2+, and 3+ represent increasing severity of cancerous lesions. Cases 1 to 4 correspond to HER2 scores of 0, 1+, 2+, and 3+, respectively. Bold text indicates incorrect classifications. GT: Real IHC images.

"3+" indicate increasing severity of cancerous lesions. As shown in Table 22, classifiers trained on many existing SOTA methods struggle to accurately determine HER2 scores, especially for the "1+" and "2+" categories. In contrast, the classifier trained on MagicStain-generated images along with real IHC images accurately recognizes all categories, demonstrating that the IHC images produced by MagicStain preserve accurate HER2 expression.

# D  ADDITIONAL WSI RESULTS

To further validate the generalizability of MagicStain on whole-slide images (WSIs), we additionally evaluate it on two H&E WSIs and two FFPE WSIs. These WSIs serve as an external test set, analogous to the H&E2IHC-HER2 and FFPE2H&E datasets in Section 4. The compared methods are selected based on their performance in the quantitative and qualitative analysis in Section 4. We select the better-performing methods Pix2pixHD (Wang et al., 2018) and AI-FFPE (Ozyoruk et al., 2021) from the GAN-based methods, and pix2pix-Turbo (Parmar et al., 2024) from the diffusion-based ones. For the H&E WSIs, all compared methods (including ours) use models trained on the H&E2IHC-HER2 dataset for inference; for the FFPE WSIs, all methods use models trained on the FFPE2H&E dataset.

Because WSIs are ultra-high-resolution and direct inference on an entire WSI is infeasible, we partition each WSI into non-overlapping patches of $1024 \times 1024$ pixels. We perform inference on each patch and then stitch the patch outputs back together to reconstruct a full WSI. Due to the absence of paired target WSIs for metric computation (e.g., FID), we present the virtually stained WSIs to a board-certified pathologist for an overall, blind ranking on a scale of 1 (best) to 4 (worst), considering both image quality and pathology correctness.

The virtually stained WSIs are shown in Fig. 6, 7, 8 and 9. The mean ranking of the four WSIs is: MagicStain (1.0), pix2pix-Turbo (2.0), Pix2pixHD (3.0), and AI-FFPE (4.0), demonstrating MagicStain's superior virtual staining quality when scaled up.

According to the pathologist's feedback, among all methods, MagicStain produced the most stable and realistic virtual staining results. More importantly, MagicStain achieves high visual quality with staining colors closely matching real HER2-IHC and H&E images, while preserving critical structure and pathology in the source input images. Although a mild mosaic ("checkerboard") artifact (Sun et al., 2023) is occasionally observed—mostly in background regions, it does not affect diagnostic interpretation. In contrast, all compared methods exhibit suboptimal staining with notable

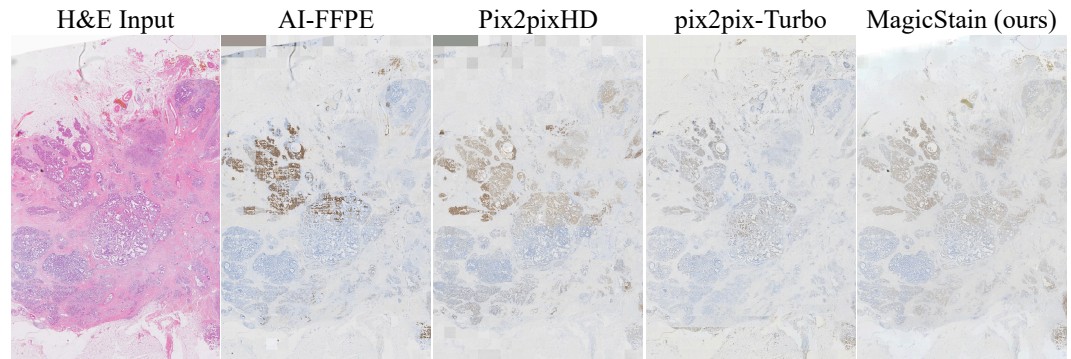

Figure 6: Virtual IHC WSIs stained from an H&E WSI input. Resolution: 19075×30000 pixels.

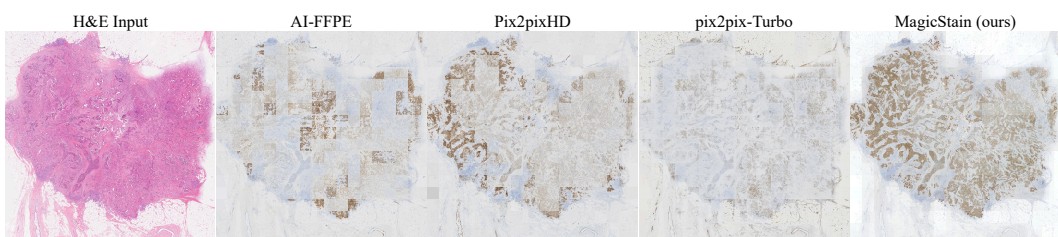

Figure 7: Virtual IHC WSIs stained from an H&E WSI input. Resolution: 18000×18000 pixels.

deficiencies, such as poor tissue structure continuity, blurred tumor cell boundaries, and incorrect cell pathology statuses—issues that substantially hinder accurate pathological assessment.

## E LIMITATIONS AND FUTURE WORK

In practical clinical settings, collecting paired cross-stain training data is typically time-consuming and costly, potentially restricting the scalability and broader adoption of MagicStain. Although paired data enhances virtual staining quality through direct correspondence, unpaired data offers much greater scalability due to its significantly larger availability. We plan to expand MagicStain by incorporating methods like CycleGAN-Turbo (Parmar et al., 2024) in future work, which allow diffusion models to be trained using unpaired data.

Compared to VIMs, our MagicStain demonstrates effective virtual staining at a high resolution of 1024×1024 pixels, taking a step further toward the efficient re-staining of large-scale images, such as WSIs. Currently, due to the megasize of WSIs and hardware constraints, directly generating an entire WSI is infeasible for almost all approaches to virtual staining. Existing methods often bypass this challenge by dividing a WSI into smaller patches, processing each patch individually, and stitching the outputs together (Asaf et al., 2024; Koivukoski et al., 2023; Rana et al., 2018; Jewsbury et al., 2024; Pati et al., 2024). However, this workaround often introduces noticeable "tiling" artifacts (Sun et al., 2023). In the future, we plan to realize tiling-artifact-free WSI virtual

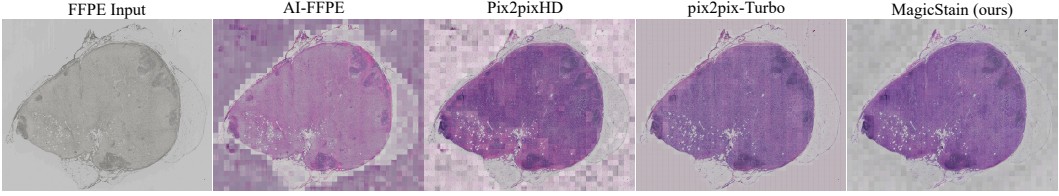

Figure 8: Virtual H&E WSIs stained from an FFPE WSI input. Resolution: 40000×32496 pixels.

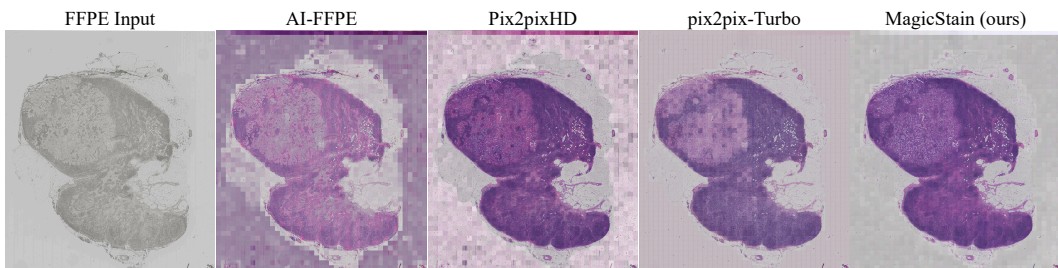

Figure 9: Virtual H&E WSIs stained from an FFPE WSI input. Resolution: 44000×51200 pixels.

staining with MagicStain, by exploring diffusion path fusion techniques such as MultiDiffusion (Bar-Tal et al., 2023).

## F   LLM USAGE DISCLOSURE

The authors affirm that no large language models (LLMs) were used in the conceptualization, ideation, experimental design, analysis, or writing of this manuscript. All theoretical development, experimental design, data analysis, and manuscript text were produced by the authors without substantive assistance from LLM-based tools. Any minor editorial proofreading or grammar checks were performed using standard non-LLM tools (e.g., spell-checkers), and all scientific content and claims remain the sole responsibility of the authors.

