# OpenReview forum: "MagicStain: High-Fidelity Pathology Image Virtual Staining via Guided Single-Step Diffusion"
_ICLR.cc/2026/Conference — Submitted to ICLR 2026_

### Official Review · Reviewer_JMPk · 2025-10-21

**Soundness:** 3
**Presentation:** 3
**Contribution:** 2
**Rating:** 2
**Confidence:** 5

**Summary:**

The authors propose a new methodology for generating virtual stains conditioned on either H&E or frozen FFPE images. The approach distills multi-step diffusion models into a single-step variant to increase inference speed, incorporates pathology-encoded information as a prior using PathClip embeddings, and introduces a hematoxylin channel loss to enforce structural constraints during generation. The model is evaluated on three different datasets with relevant ablation studies.

**Strengths:**

In my opinion, the paper has several strengths:

1. The use of diffusion distillation methods to improve convergence and generation efficiency is both technically sound and impactful.

2. The evaluation across three different datasets, including an out-of-distribution dataset without any fine-tuning, is particularly interesting and relevant.

**Weaknesses:**

There are several weaknesses of the papers:

1. First is novelty. I have some concerns about novelty of the proposed methods and Choice of losses.

(a) Using histopathology pre-trained models for generation has been explored before [1,2], perhaps with different encoders, and has been shown to perform well. Therefore, claiming the use of PathClip embeddings as a novelty seems questionable. Additionally, the paper does not experimentally or textually justify why PathClip would outperform vision-only encoders used in previous works such as UNI or Virchow. What is the specific reason that PathClip embeddings should improve generation? Is there evidence that these embeddings provide a tangible advantage for virtual staining tasks?

(b) Although the H-channel loss has been used before[3], it seems primarily relevant for FFPE-to-H&E staining, where the goal is to accurately reconstruct the hematoxylin channel. For other tasks, such as H&E-to-IHC translation, the focus should be on the DAB channel rather than the H-channel. If the model emphasizes the H-channel too strongly, it could produce inaccurate results for the DAB channel, which is undesirable. Therefore, the rationale for adding this loss in the broader context appears flawed.

(c) Progressive training may be useful for models with slow inference times, but the proposed model is single-step, so its relevance is unclear. Reducing WSI generation time from five minutes to two or three minutes is unlikely to be significant for pathology applications. Progressive training would be more justified for multi-step diffusion models, where it could reduce inference times from hours to minutes. Therefore, I find the rationale for applying progressive training here neither logically sound nor practically relevant.


(2). No reasoning is provided for using the Wilcoxon signed-rank test. Why not a simple t-test? Which assumption of the t-test is violated here that necessitates a different statistical test?

(3) No relevant downstream application is provided. The proposed downstream task is unfamiliar to me— is it a standard in the field, or are the authors proposing a new task? If it is new, they should provide references to support its validity. If not, it is unclear how we can trust the results of the downstream evaluation.

(4) Manual evaluation is performed only on H&E WSI images; no manual evaluation on IHC images is reported, even in the appendix.


[1] Ho, Man M., Shikha Dubey, Yosep Chong, Beatrice Knudsen, and Tolga Tasdizen. "F2fldm: Latent diffusion models with histopathology pre-trained embeddings for unpaired frozen section to ffpe translation." In 2025 IEEE/CVF Winter Conference on Applications of Computer Vision (WACV), pp. 4382-4391. IEEE, 2025.

[2] Yang, Hao, JianYu Wu, Run Fang, Xuelian Zhao, Yuan Ji, Zhiyu Chen, Guibin He, Junceng Guo, Yang Liu, and Xinhua Zeng. "Cross-channel Perception Learning for H&E-to-IHC Virtual Staining." arXiv preprint arXiv:2506.07559 (2025).

[3] Peng, Qiong, Weiping Lin, Yihuang Hu, Ailisi Bao, Chenyu Lian, Weiwei Wei, Meng Yue, Jingxin Liu, Lequan Yu, and Liansheng Wang. "Advancing H&E-to-IHC virtual staining with task-specific domain knowledge for HER2 scoring." In International Conference on Medical Image Computing and Computer-Assisted Intervention, pp. 3-13. Cham: Springer Nature Switzerland, 2024.

**Questions:**

I have several question some of which i stated in the Weaknesses above, other questions and suggestions that i have are:

1. You have reported that models are trained for 15-20 epochs. But while reading other papers, pix2pix and CycleGAN models are generally trained for 200 epochs, why this discrepancy ?

2. Comparisons to [1] and [2] are missing, even though both were trained under the same settings as the proposed method. Other papers in the literature also achieve reasonable performance, making such comparisons important.

3. Do you plan to release the datasets used in this study publicly? If not, how can others access the data or verify that the results are reproducible? As no public datasets are used in the paper, reproducibility will be hard if dataset access is limited. Or you need to use at-least one public dataset.


[1] Li, Fangda, Zhiqiang Hu, Wen Chen, and Avinash Kak. "Adaptive supervised patchnce loss for learning h&e-to-ihc stain translation with inconsistent groundtruth image pairs." In International Conference on Medical Image Computing and Computer-Assisted Intervention, pp. 632-641. Cham: Springer Nature Switzerland, 2023.

[2] Liu, Shengjie, Chuang Zhu, Feng Xu, Xinyu Jia, Zhongyue Shi, and Mulan Jin. "Bci: Breast cancer immunohistochemical image generation through pyramid pix2pix." In Proceedings of the IEEE/CVF conference on computer vision and pattern recognition, pp. 1815-1824. 2022.

---

> ### Author Response · Authors · 2025-11-23
> **Response to Reviewer JMPk (Part 1/5)**
>
> > **W1**: Using histopathology pre-trained models for generation has been explored before [1,2], perhaps with different encoders, and has been shown to perform well. Therefore, claiming the use of PathCLIP embeddings as a novelty seems questionable. Additionally, the paper does not experimentally or textually justify why PathCLIP would outperform vision-only encoders used in previous works such as UNI or Virchow. What is the specific reason that PathCLIP embeddings should improve generation? Is there evidence that these embeddings provide a tangible advantage for virtual staining tasks?
>
> **A1**: We appreciate the reviewer's valuable comment.
>
> [1] employs histopathology pretrained models based on the DINO architecture, trained via self-supervised learning, to assist virtual staining in diffusion models; however, it does not explore using embeddings extracted by histopathology pretrained models as a loss supervision signal.
> [2] uses Gigapath, a DINO-based histopathology pretrained models, to extract image embeddings and imposes a similarity-preserving loss.
> However, [2] optimizes a GAN-based model and does not explore diffusion architectures.
> **We have included discussions of [1, 2] in Section "2 RELATED WORK".**
>
> &emsp;
> As noted in the “INTRODUCTION,” the baseline pix2pix-Turbo, as done in VIMs, fails to correctly transfer pathology from source images to the target domain due to limited domain expertise.
> Therefore, we are strongly motivated to apply histopathology pretrained models to compensate for pathology domain expertise in diffusion-based virtual staining, which represents an application-level novelty.
> Compared with [1, 2], we comprehensively use histopathology pretrained models: we incorporate both loss supervision and pathology-prior embedding guidance, and we apply them to single-step diffusion models, which have not previously been explored for virtual staining.
>
> &emsp;
> PathCLIP, a pretrained vision-language model (VLM) specialized for pathology, is trained via contrastive learning between textual and visual embeddings.
> This distinguishes it from DINO-based histopathology pretrained models: PathCLIP carries semantic priors aligned with human textual descriptions, which can provide more appropriate cues for understanding pathology in the source domain.
> To justify that PathCLIP outperforms vision-only encoders, we replace PathCLIP with UNI2-h or Virchow2 on the H&E2IHC-HER2 dataset for both I2I guidance and pathology-accuracy supervision.
> As shown in the table below, PathCLIP achieves better performance, likely because its human-aligned semantic priors provide more suitable pathology-relevant information than DINO-based histopathology pretrained models.
>
> | Method | FID↓ | KID↓ | PSNR↑ | SSIM↑ |
> |-------|-------|-------|-------|-------|
> | UNI2-h | 33.84 | 0.0064 | 18.50±3.721 | 0.408±0.126 |
> | Virchow2 | 33.94 | 0.0050 | 18.32±3.624 | 0.402±0.127 |
> | PathCLIP | 33.51 | 0.0033 | 18.58±3.668 | 0.411±0.125 |
>
> Despite that, using UNI2-h or Virchow2 yields performance competitive with leading methods in Table 1, which suggests that **MagicStain can also effectively incorporate other histopathology pretrained models to achieve consistently strong performance**.
> **We have added these results in Appendix “C.2 CHOICE OF PATHOLOGY PRETRAINED MODEL.”**
>
> &emsp;
> Although using PathCLIP embeddings may seem straightforward, they are demonstrably effective.
> We ablate the contributions of PathCLIP priors—including guidance for I2I translation (w/o prior) and pathology-accuracy supervision (w/o $L_{Expert}$)—as reported in Table 3.
> Removing either component results in **pronounced drops** in the corresponding metrics, demonstrating the substantial benefits of PathCLIP guidance in MagicStain.
>
> - [1] Ho, Man M., Shikha Dubey, Yosep Chong, Beatrice Knudsen, and Tolga Tasdizen. "F2fldm: Latent diffusion models with histopathology pre-trained embeddings for unpaired frozen section to FFPE translation." In 2025 IEEE/CVF Winter Conference on Applications of Computer Vision (WACV), pp. 4382-4391. IEEE, 2025.
> - [2] Yang, Hao, JianYu Wu, Run Fang, Xuelian Zhao, Yuan Ji, Zhiyu Chen, Guibin He, Junceng Guo, Yang Liu, and Xinhua Zeng. "Cross-channel Perception Learning for H&E-to-IHC Virtual Staining." arXiv preprint arXiv:2506.07559 (2025).

---

> ### Author Response · Authors · 2025-11-23
> **Response to Reviewer JMPk (Part 2/5)**
>
> > **W2**: Although the H-channel loss has been used before[3], it seems primarily relevant for FFPE-to-H&E staining, where the goal is to accurately reconstruct the hematoxylin channel. For other tasks, such as H&E-to-IHC translation, the focus should be on the DAB channel rather than the H-channel. If the model emphasizes the H-channel too strongly, it could produce inaccurate results for the DAB channel, which is undesirable. Therefore, the rationale for adding this loss in the broader context appears flawed.
>
> **A2**: Thank you for the insightful questions.
>
> The brown DAB channel in IHC images highlights the location and intensity of target protein expression (e.g., HER2 on the cell membrane and CK in epithelial cells; darker brown indicates stronger expression), which primarily encodes pathological information.
> MagicStain relies on PathCLIP to supervise pathological accuracy when generating H&E or IHC images, which has been demonstrated to be effective in Table 3, serving a similar purpose to a loss that emphasizes the DAB channel.
> Also, MagicStain incorporates **global supervision** across the entire image (Hematoxylin (H) channel or DAB channel) through an **MSE loss in the latent space** and a **LPIPS loss in the pixel space**, which also ensures the accuracy of the DAB channel.
>
> &emsp;
> After reading the reviewer's comments, we have conducted experiments on the H&E2IHC-HER2 dataset by adding perceptual and reconstruction losses on the DAB-channel instead of the H-channel (w DAB w/o H) or by using both (w DAB w H).
> The results below show that jointly enhancing the DAB- and H-channels yields performance similar to not enhancing the DAB-channel (slightly better FID and PSNR, and slightly worse KID and SSIM).
>
> | Method | FID↓ | KID↓ | PSNR↑ | SSIM↑ |
> |-------|-------|-------|-------|-------|
> | w DAB w H | 33.18 | 0.0049 | 18.60±3.675 | 0.405±0.126 |
> | w DAB w/o H | 35.72 | 0.0086 | 18.31±3.597 | 0.403±0.125 |
> | w/o DAB w H (Ours) | 33.51 | 0.0033 | 18.58±3.668 | 0.411±0.125 |
>
> However, **removing H-channel enhancement** (w DAB w/o H) results in clearly inferior performance, indicating that **H-channel enhancement is crucial** for MagicStain.
> **We have added these results in Appendix “C.9 IMPACT OF DAB-CHANNEL ENHANCEMENT.”**
>
> &emsp;
> Furthermore, the **DAB-channel loss** can be used **only** for generating IHC images and cannot be applied to FFPE-to-H&E staining, limiting the method’s applicability across multiple stain types.
> In contrast, the H-channel, which highlights nuclei in both H&E and IHC images, can be used for both generation tasks.
> Therefore, MagicStain enhances the fidelity of pathological and nuclear morphology by imposing H-channel loss and PathCLIP similarity loss for H&E and IHC image generation.
>
> > **W3**: Progressive training may be useful for models with slow inference times, but the proposed model is single-step, so its relevance is unclear. Reducing WSI generation time from five minutes to two or three minutes is unlikely to be significant for pathology applications. Progressive training would be more justified for multi-step diffusion models, where it could reduce inference times from hours to minutes. Therefore, I find the rationale for applying progressive training here neither logically sound nor practically relevant.
>
> **A3**: We appreciate the reviewer's insightful comments.
>
> As noted, **progressive training** stabilizes learning by starting with easier samples or tasks and gradually introducing more complex data or objectives.
> For accelerating multi-step diffusion models, one typically trains the model with many steps, high precision, and large parameters (easy tasks), then applies step distillation, quantization, pruning, etc., to maintain accuracy with fewer steps, lower precision, and fewer parameters (hard tasks).
> In this work, although MagicStain is already single-step, it still faces the **challenge of high-resolution adaptation**.
> Therefore, we first train the model on **easier, low-resolution images** for rapid convergence, and then fine-tune it on **more challenging high-resolution images** using LoRA.
> Experiments in Table 3 and Section "Ablation Studies" demonstrate that progressive training enables fast generalization from low to high resolutions, and achieves better performance than directly training a high-resolution model from scratch.
>
> - [3] Peng, Qiong, Weiping Lin, Yihuang Hu, Ailisi Bao, Chenyu Lian, Weiwei Wei, Meng Yue, Jingxin Liu, Lequan Yu, and Liansheng Wang. "Advancing H&E-to-IHC virtual staining with task-specific domain knowledge for HER2 scoring." In International Conference on Medical Image Computing and Computer-Assisted Intervention, pp. 3-13. Cham: Springer Nature Switzerland, 2024.

---

> ### Author Response · Authors · 2025-11-23
> **Response to Reviewer JMPk (Part 3/5)**
>
> > **W4**: No reasoning is provided for using the Wilcoxon signed-rank test. Why not a simple t-test? Which assumption of the t-test is violated here that necessitates a different statistical test?
>
> **A4**: Thank you for the insightful questions.
>
> Although simple, a t-test requires three assumptions: independence, normality, and homogeneity of variances.
> However, many of the comparisons in our paper fail to meet the assumptions of normality or homogeneity of variance.
> To use a consistent test across comparisons, we opt for the less restricted Wilcoxon signed-rank test.
>
>
> > **W5**: No relevant downstream application is provided. The proposed downstream task is unfamiliar to me— is it a standard in the field, or are the authors proposing a new task? If it is new, they should provide references to support its validity. If not, it is unclear how we can trust the results of the downstream evaluation.
>
> **A5**: We appreciate the reviewer's constructive comment.
>
> Our downstream task follows the same protocol as in [3], as described in "Evaluation on Downstream Task" of Section "4 EXPERIMENTS".
> This downstream evaluation has been shown to effectively examine the practical usability of virtually stained IHC images in clinical tasks.
> Both our work and [3] train a classifier on virtually stained IHC images from the H&E2IHC-HER2 dataset, and use the classifier to predict HER2 scoring of the real IHC images in the H&E2IHC-Ext dataset.
> As shown in Table 5, the classifier trained on IHC images virtually stained by MagicStain outperforms those trained on images by all compared methods—and is even competitive with the upper-bound classifier trained on real IHC images.
> This indicates that MagicStain-generated IHC images exhibit accurate HER2 expression. **We have already stated in "Evaluation on Downstream Task" that this setup follows [3], supporting the validity of our downstream evaluation.**
>
> > **W6**: Manual evaluation is performed only on H&E WSI images; no manual evaluation on IHC images is reported, even in the appendix.
>
> **A6**: We appreciate the reviewer's insightful comments.
>
> We respectfully clarify that, in addition to the virtual H&E WSIs in Figures 7 and 8, we also provide a qualitative analysis of virtual IHC WSIs in Figures 5 and 6, along with the pathologist’s evaluation in APPENDIX “D ADDITIONAL WSI RESULTS”.
> According to expert feedback, MagicStain produces the most stable and realistic virtual staining results among all compared methods.
> It achieves high visual quality, with staining colors closely matching those of real HER2-IHC and H&E images, while preserving critical structure and pathology in the source images.
> Although a mild mosaic (“checkerboard”) artifact is occasionally observed—mostly in background regions, it does not affect diagnostic interpretation.
>
> > **Q1**: You have reported that models are trained for 15-20 epochs. But while reading other papers, pix2pix and CycleGAN models are generally trained for 200 epochs, why this discrepancy ?
>
> **A7**: We thank the reviewer for the insightful question.
>
> MagicStain is a simple yet effective single-step diffusion model that integrates domain-specific modules, medically motivated loss functions, and a progressive training strategy for high-resolution adaptation.
> These designs enable the model to converge within only **15–20 epochs**, achieving high-fidelity, high-resolution virtual staining and surpassing pix2pix and CycleGAN models that require **200 epochs** of training, as demonstrated in "4 EXPERIMENTS" on both virtual staining and downstream classification tasks.
>
> &emsp;
> We also trained MagicStain for 200 epochs (150 for stage 1 and 50 for stage 2), following the same ratio as in the main text (15 and 5 epochs).
> As shown below, the 200-epoch model yields results comparable to those obtained with 20 epochs, indicating that MagicStain has already converged at 20 epochs.
> However, longer training significantly increases computational and energy costs.
> Thus, we consider **20 epochs** sufficient.
> **We have added these results in Appendix “C.8 VERIFYING TRAINING CONVERGENCE.”**
>
> | Epochs | FID↓ | KID↓ | PSNR↑ | SSIM↑ |
> |-------|-------|-------|-------|-------|
> | 200 | 31.76 | 0.0060 | 18.55±3.754 | 0.415±0.128 |
> | 20 | 33.51 | 0.0033 | 18.58±3.668 | 0.411±0.125 |
>
> - [3] Peng, Qiong, Weiping Lin, Yihuang Hu, Ailisi Bao, Chenyu Lian, Weiwei Wei, Meng Yue, Jingxin Liu, Lequan Yu, and Liansheng Wang. "Advancing H&E-to-IHC virtual staining with task-specific domain knowledge for HER2 scoring." In International Conference on Medical Image Computing and Computer-Assisted Intervention, pp. 3-13. Cham: Springer Nature Switzerland, 2024.

---

> ### Author Response · Authors · 2025-11-23
> **Response to Reviewer JMPk (Part 4/5)**
>
> > **Q2**: Comparisons to [4] and [5] are missing, even though both were trained under the same settings as the proposed method. Other papers in the literature also achieve reasonable performance, making such comparisons important.
>
> **A8**: We appreciate the reviewer's excellent suggestion.
>
> Thus, we compare with ASP [4] and Pyramidpix2pix [5] on the H&E2IHC-HER2 dataset below.
> MagicStain achieves better performance across all metrics than ASP [4] and Pyramidpix2pix [5].
> **We have added these new comparisons in Appendix “C.7 ADDITIONAL COMPARISON WITH EXISTING METHODS.”**
>
> | Method | FID↓ | KID↓ | PSNR↑ | SSIM↑ |
> |-------|-------|-------|-------|-------|
> | ASP [4] | 65.51 | 0.0294 | 17.30±2.966 | 0.368±0.104 |
> | Pyramidpix2pix [5] | 47.52 | 0.0183 | 18.21±3.722 | 0.389±0.124 |
> | MagicStain (Ours) | 33.51 | 0.0033 | 18.58±3.668 | 0.411±0.125 |
>
> > **Q3**: Do you plan to release the datasets used in this study publicly? If not, how can others access the data or verify that the results are reproducible? As no public datasets are used in the paper, reproducibility will be hard if dataset access is limited. Or you need to use at-least one public dataset.
>
> **A9**: We appreciate the reviewer's excellent suggestion.
>
> Due to objective constraints, we are unable to release H&E2IHC-HER2, H&E2IHC-CK, and FFPE2H&E at this time.
> However, following the reviewer's suggestion, we have conducted **additional experiments on widely used public datasets**, including **BCI [5]** and the four **MIST [4]** datasets (HER2, ER, Ki67, PR), and compared MagicStain with SOTA approaches under the same settings as in Table 1.
>
> ### BCI
> | Method | FID↓ | KID↓ | PSNR↑ | SSIM↑ |
> |-------|-------|-------|-------|-------|
> | Pix2pix | 91.06 | 0.0580 | 21.28±4.773 | 0.544±0.172 |
> | Pix2pixHD | 88.81 | 0.0583 | 19.47±5.657 | 0.532±0.172 |
> | [3] | 63.18 | 0.0258 | 19.96±5.263 | 0.517±0.169 |
> | CycleGAN | 61.09 | 0.0304 | 16.46±5.858 | 0.511±0.166 |
> | AI-FFPE | 51.31 | 0.0226 | 15.09±4.837 | 0.491±0.147 |
> | ControlNet | 236.9 | 0.2381 | 9.100±2.864 | 0.248±0.065 |
> | StainFuser | 80.49 | 0.0519 | 13.54±4.243 | 0.519±0.174 |
> | pix2pix-Turbo | 303.1 | 0.4019 | 13.64±4.494 | 0.443±0.160 |
> | MagicStain (ours) | **30.24** | **0.0037** | **22.38**±4.252 | **0.550**±0.183 |
>
> ### MIST-HER2
> | Method | FID↓ | KID↓ | PSNR↑ | SSIM↑ |
> |-------|-------|-------|-------|-------|
> | Pix2pix | 75.97 | 0.0427 | 15.02±2.855 | 0.249±0.084 |
> | Pix2pixHD | 61.45 | 0.0236 | 14.59±2.967 | 0.243±0.085 |
> | [3] | 50.95 | 0.0148 | 14.58±2.971 | 0.241±0.087 |
> | CycleGAN | 67.78 | 0.0343 | 13.91±2.621 | 0.228±0.077 |
> | AI-FFPE | 57.41 | 0.0217 | 13.37±2.433 | 0.225±0.076 |
> | ControlNet | 170.2 | 0.1244 | 8.335±1.394 | 0.135±0.024 |
> | StainFuser | 88.51 | 0.0490 | 11.11±1.819 | 0.254±0.092 |
> | pix2pix-Turbo | 152.5 | 0.1018 | 14.04±2.634 | 0.214±0.072 |
> | MagicStain (ours) | **36.71** | **0.0049** | **15.12**±2.917 | **0.257**±0.093 |
>
> ### MIST-ER
> | Method | FID↓ | KID↓ | PSNR↑ | SSIM↑ |
> |-------|-------|-------|-------|-------|
> | Pix2pix | 70.53 | 0.0382 | 15.071±3.209 | 0.269±0.093 |
> | Pix2pixHD | 58.45 | 0.0213 | 14.468±3.192 | 0.263±0.095 |
> | [3] | 54.68 | 0.0204 | 14.352±3.164 | 0.255±0.093 |
> | CycleGAN | 46.66 | 0.0142 | 13.113±2.516 | 0.247±0.093 |
> | AI-FFPE | 72.54 | 0.0413 | 14.065±2.738 | 0.254±0.080 |
> | ControlNet | 245.6 | 0.2156 | 7.3965±1.209 | 0.124±0.020 |
> | StainFuser | 98.33 | 0.0615 | 10.094±1.168 | 0.263±0.099 |
> | pix2pix-Turbo | 79.73 | 0.0371 | 13.458±2.406 | 0.209±0.079 |
> | MagicStain (ours) | **30.52** | **0.0032** | **15.074**±3.287 | **0.275**±0.103 |
>
> - [3] Peng, Qiong, Weiping Lin, Yihuang Hu, Ailisi Bao, Chenyu Lian, Weiwei Wei, Meng Yue, Jingxin Liu, Lequan Yu, and Liansheng Wang. "Advancing H&E-to-IHC virtual staining with task-specific domain knowledge for HER2 scoring." In International Conference on Medical Image Computing and Computer-Assisted Intervention, pp. 3-13. Cham: Springer Nature Switzerland, 2024.
> - [4] Li, Fangda, Zhiqiang Hu, Wen Chen, and Avinash Kak. "Adaptive supervised patchnce loss for learning h&e-to-ihc stain translation with inconsistent groundtruth image pairs." In International Conference on Medical Image Computing and Computer-Assisted Intervention, pp. 632-641. Cham: Springer Nature Switzerland, 2023.
> - [5] Liu, Shengjie, Chuang Zhu, Feng Xu, Xinyu Jia, Zhongyue Shi, and Mulan Jin. "Bci: Breast cancer immunohistochemical image generation through pyramid pix2pix." In Proceedings of the IEEE/CVF conference on computer vision and pattern recognition, pp. 1815-1824. 2022.

---

> ### Author Response · Authors · 2025-11-23
> **Response to Reviewer JMPk (Part 5/5)**
>
> ### MIST-Ki67
> | Method | FID↓ | KID↓ | PSNR↑ | SSIM↑ |
> |-------|-------|-------|-------|-------|
> | Pix2pix | 64.99 | 0.0441 | 15.21±2.146 | 0.290±0.101 |
> | Pix2pixHD | 53.29 | 0.0331 | 14.59±2.162 | 0.284±0.100 |
> | [3] | 44.95 | 0.0223 | 14.64±2.184 | 0.278±0.100 |
> | CycleGAN | 38.77 | 0.0122 | 13.70±1.908 | 0.252±0.089 |
> | AI-FFPE | 36.98 | 0.0090 | 14.04±1.946 | 0.275±0.086 |
> | ControlNet | 250.3 | 0.2851 | 8.272±1.324 | 0.170±0.031 |
> | StainFuser | 66.67 | 0.0408 | 10.97±1.307 | **0.312**±0.100 |
> | pix2pix-Turbo | 161.5 | 0.1529 | 13.49±1.458 | 0.232±0.080 |
> | MagicStain (ours) | **26.13** | **0.0034** | **15.25**±2.199 | 0.297±0.105 |
>
> ### MIST-PR
> | Method | FID↓ | KID↓ | PSNR↑ | SSIM↑ |
> |-------|-------|-------|-------|-------|
> | Pix2pix | 75.11 | 0.0420 | **15.41**±3.534 | 0.286±0.105 |
> | Pix2pixHD | 60.08 | 0.0239 | 14.79±3.563 | 0.276±0.106 |
> | [3] | 65.80 | 0.0323 | 14.82±3.520 | 0.262±0.100 |
> | CycleGAN | 45.69 | 0.0113 | 13.40±2.919 | 0.239±0.086 |
> | AI-FFPE | 49.93 | 0.0193 | 13.48±3.060 | 0.256±0.089 |
> | ControlNet | 272.9 | 0.2609 | 7.585±1.162 | 0.148±0.028 |
> | StainFuser | 80.15 | 0.0461 | 10.26±1.530 | 0.279±0.124 |
> | pix2pix-Turbo | 78.62 | 0.0436 | 14.89±3.126 | 0.259±0.095 |
> | MagicStain (ours) | **29.94** | **0.0030** |15.36±3.540 | **0.290**±0.113 |
>
>
> As shown above, MagicStain achieves the **best performance across most of metrics on the five public datasets** (only little lower PSNR compared with Pix2pix on MIST-PR and SSIM compared with StainFuser on MIST-Ki67), demonstrating superior perceptual and structural quality in its virtually stained images as well as excellent generalization across datasets.
> **We have added these results in Appendix “C.6 ADDITIONAL EXPERIMENTS ON PUBLIC DATASETS.”**
> In addition, our code and trained models will be released to facilitate reproducible research and further exploration in this direction by the community.
>
> - [3] Peng, Qiong, Weiping Lin, Yihuang Hu, Ailisi Bao, Chenyu Lian, Weiwei Wei, Meng Yue, Jingxin Liu, Lequan Yu, and Liansheng Wang. "Advancing H&E-to-IHC virtual staining with task-specific domain knowledge for HER2 scoring." In International Conference on Medical Image Computing and Computer-Assisted Intervention, pp. 3-13. Cham: Springer Nature Switzerland, 2024.

---

> ### Comment · Reviewer_JMPk · 2025-11-26
> **Response to Authors Rebuttal**
>
> **I thank the reviewer for considering my suggestion for improving the impact of the paper. I will increase the score to acknowledge the authors’ detailed rebuttal; however, this increase is modest, as I still have concerns regarding the novelty of the work and authors arguments about each point:**
>
> > 1.The performance differences between the encoder variants are negligible under PSNR and SSIM. Moreover, FID and KID can be misleading when evaluating the accuracy of virtual staining, and improvements in these metrics alone should not be over-interpreted Especially in Medical Image Translation. While a recent studies [1,2,3,4] supports this point, its applicability here is uncertain given potential differences in dataset acquisition between your study and theirs, which may limit the relevance of its findings. A similar line of reasoning applies to the comparison with and without the H-loss objective. Overall, I am not fully convinced by the arguments put forward regarding the impact of the encoder choices or the importance of the H-loss in this context.
>
> > 2. I am not sure how to interpret the downstream task results in [3] or those used in this paper, as the evaluation code has not been released. Consequently, it is unclear what exactly is being measured. Additionally, the paper does not provide qualitative examples—such as the classification boundary or representative samples—to help contextualize the results. Including such qualitative visualizations would significantly improve clarity.
>
> [1] Kataria, Tushar, Shikha Dubey, Mary Bronner, Jolanta Jedrzkiewicz, Ben J. Brintz, Shireen Y. Elhabian, and Beatrice S. Knudsen. "Building Trust in Virtual Immunohistochemistry: Automated Assessment of Image Quality." arXiv preprint arXiv:2511.04615 (2025).
>
> [2] Deo, Yash, Yan Jia, Toni Lassila, William AP Smith, Tom Lawton, Siyuan Kang, Alejandro F. Frangi, and Ibrahim Habli. "Metrics that matter: Evaluating image quality metrics for medical image generation." arXiv preprint arXiv:2505.07175 (2025).
>
> [3] Dohmen, Melanie, Mark A. Klemens, Ivo M. Baltruschat, Tuan Truong, and Matthias Lenga. "Similarity and quality metrics for MR image-to-image translation." Scientific Reports 15, no. 1 (2025): 3853.
>
> [4] Dohmen, Melanie, Mark A. Klemens, Ivo M. Baltruschat, Tuan Truong, and Matthias Lenga. "Similarity and quality metrics for MR image-to-image translation." Scientific Reports 15, no. 1 (2025): 3853.

---

> ### Author Response · Authors · 2025-12-03
> **Response to Reviewer JMPk (Part 1/2)**
>
> Thank you for your appreciation of our detailed rebuttal.
>
> > **Q4**: 1.The performance differences between the encoder variants are negligible under PSNR and SSIM. Moreover, FID and KID can be misleading when evaluating the accuracy of virtual staining, and improvements in these metrics alone should not be over-interpreted Especially in Medical Image Translation. While a recent studies supports this point, its applicability here is uncertain given potential differences in dataset acquisition between your study and theirs, which may limit the relevance of its findings. A similar line of reasoning applies to the comparison with and without the H-loss objective. Overall, I am not fully convinced by the arguments put forward regarding the impact of the encoder choices or the importance of the H-loss in this context.
>
> **A10**: We appreciate the reviewer's insightful comments.
> We acknowledge the concern regarding “Using FID and KID to explain the accuracy of virtual staining may be misleading,” and have used segmentation metrics to evaluate the impact of the encoder choices and H-channel losses in this context.
> Following the methodology in [1], we deconvolute IHC images into hematoxylin and DAB, a brown chromogen (i.e., IHC-positive), and compare the DAB masks between real and virtual images.
> Specifically, we compute the DICE score, intersection over union (IoU), true positive rate (TPR), and true negative rate (TNR) on the H&E2IHC-HER2 dataset.
> Higher values of these metrics reflect better alignment of DAB-mask between real and virtual images, indicating improved pathological accuracy.
> We conduct the Wilcoxon signed-rank test for pairwise comparison with our method on each metric value, using “*” to denote $p < 0.05$ in the table.
>
> | Method | TPR↑ | TNR↑ | DICE↑ | IoU↑ |
> |-------|-------|-------|-------|-------|
> | w/o prior | 0.3579* | 0.8462* | 0.3271* | 0.2523* |
> | w/o $L_{Expert}$ | 0.3587* | 0.8444* | 0.3248* | 0.2498* |
> | UNI2-h | 0.3663* | 0.8523* | 0.3306* | 0.2533* |
> | Virchow2 | 0.3667* | 0.8546* | 0.3325* | 0.2569* |
> | w/o $L_{MSE}$ | 0.3593* | 0.8446* | 0.3245* | 0.2499* |
> | w/o $L_{LPIPS}$ | 0.3492* | 0.8425* | 0.3283* | 0.2492* |
> | w/o $L^{H}_{MSE}$ | 0.3733 | 0.8525* | 0.3320* | 0.2556* |
> | w/o $L^{H}_{LPIPS}$ | 0.3722 | 0.8598 | 0.3334 | 0.2558 |
> | w DAB w H | 0.3775 | 0.8546* | 0.3347 | 0.2584 |
> | w DAB w/o H | 0.3743 | 0.8550* | 0.3369 | 0.2582 |
> | w/o DAB w H (Ours) | 0.3717 | 0.8567 | 0.3346 | 0.2590 |
>
> &emsp;
> As shown above, removing the pathology prior (“w/o prior”) or its associated loss (“w/o $L_{Expert}$”), which enforces pathology-related consistency, degrades the pathological accuracy of the generated results.
> Replacing PathCLIP with UNI2-h or Virchow2 for both I2I guidance and pathology-accuracy supervision leads to slight drops across all metrics, demonstrating that the pretrained pathology VLM (PathCLIP) provides more appropriate pathology-relevant priors than DINO-based histopathology pretrained models (UNI2-h and Virchow2).
> Removing losses imposed on the entire image—MSE and LPIPS (“w/o $L_{MSE}$” and “w/o $L_{LPIPS}$”) also harms performance.
>
> &emsp;
> Removing the corresponding losses on the H-channel ("w/o $L^{H}\_{MSE}$" and "w/o $L^{H}\_{LPIPS}$") yields results comparable to those obtained when including them (“w/o DAB w H (Ours)”), indicating that the H-channel losses do **not, contrary to the reviewer’s concern, lead to inaccurate DAB-channel outputs** (“If the model emphasizes the H-channel too strongly, it could produce inaccurate results for the DAB channel, which is undesirable.”).
> However, as shown in Table 3, removing the MSE and LPIPS losses on the H-channel, which highlights nuclei in the target-domain images, results in pronounced drops in perceptual metrics (FID and KID) and structural metrics (PSNR and SSIM).
> Meanwhile, explicitly emphasizing the DAB channel (“w DAB w H” and “w DAB w/o H”) does not yield additional gains, as PathCLIP already provides sufficient guidance.
> This demonstrates that our enhancement of the H-channel is well justified.
> Therefore, both H-channel losses, and the supervision combined with the PathCLIP prior, are essential for achieving optimal performance of MagicStain.
>
> &emsp;
> We have revised the description of FID and KID to: “lower perception-oriented metric scores indicate a smaller disparity between the distributions of generated results and the target domains.”
> We have included the DAB mask accuracy evaluations in Appendix “C.13 EVALUATION OF SEGMENTATION METRICS FOR ABLATION STUDIES.”
>
> - [1] Kataria, Tushar, Shikha Dubey, Mary Bronner, Jolanta Jedrzkiewicz, Ben J. Brintz, Shireen Y. Elhabian, and Beatrice S. Knudsen. "Building Trust in Virtual Immunohistochemistry: Automated Assessment of Image Quality." arXiv preprint arXiv:2511.04615 (2025).

---

> ### Author Response · Authors · 2025-12-03
> **Response to Reviewer JMPk (Part 2/2)**
>
> > **Q5**: I am not sure how to interpret the downstream task results in [3] or those used in this paper, as the evaluation code has not been released. Consequently, it is unclear what exactly is being measured. Additionally, the paper does not provide qualitative examples—such as the classification boundary or representative samples—to help contextualize the results. Including such qualitative visualizations would significantly improve clarity.
>
> **A11**: We appreciate the reviewer's excellent suggestion.
>
> We respectfully clarify that the downstream task results in our work can be interpreted as evidence that **MagicStain captures diagnostically relevant information during virtual staining, and its virtually stained IHC images are practically usable in downstream clinical tasks**.
> Our protocol follows the same setup as [3], with the only difference being that [3] evaluates their proposed method.
>
> &emsp;
> Our downstream task focuses on assessing HER2 expression in virtually stained IHC images.
> As described in Lines 430–448: “For each virtual staining method, ... and another one using real IHC images in the test set of H&E2IHC-HER2.”, we train a classifier on the generated IHC images of each method, on H&E images, and on real IHC images, and then evaluate all classifiers on real IHC images from the H&E2IHC-Ext dataset.
> If a method produces IHC images with more accurate HER2 expression (which also reflects better pathological consistency), then the classifier trained on its generated images will exhibit correspondingly stronger performance.
> Because real IHC images already represent practical usability in downstream clinical tasks, virtually stained IHC images of one method whose classifier achieves performance competitive with that of real IHC images can be interpreted as **possessing both clinical relevance and the potential for accurate virtual staining**.
>
> &emsp;
> We have provided qualitative examples of the downstream task, including the images to be classified in Figure 5 and the classification results of various methods in Table 22.
> In HER2 scoring, “0” denotes no cancerous lesion, while “1+”, “2+”, and “3+” indicate increasing severity of cancerous lesions.
> As shown in Table 22, classifiers trained on many existing SOTA methods struggle to accurately determine HER2 scores, especially for the “1+” and “2+” categories.
> In contrast, the classifier trained on MagicStain-generated images along with real IHC images accurately recognizes all categories, demonstrating that the IHC images produced by MagicStain preserve **accurate HER2 expression**.
>
> - [3] Peng, Qiong, Weiping Lin, Yihuang Hu, Ailisi Bao, Chenyu Lian, Weiwei Wei, Meng Yue, Jingxin Liu, Lequan Yu, and Liansheng Wang. "Advancing H&E-to-IHC virtual staining with task-specific domain knowledge for HER2 scoring." In International Conference on Medical Image Computing and Computer-Assisted Intervention, pp. 3-13. Cham: Springer Nature Switzerland, 2024.

---

### Official Review · Reviewer_6Jo9 · 2025-10-31

**Soundness:** 3
**Presentation:** 3
**Contribution:** 2
**Rating:** 4
**Confidence:** 4

**Summary:**

This paper addresses the challenge of virtual staining in histopathology. To this end, the authors propose a one-step diffusion-based framework. To mitigate the domain gap in image synthesis, a retrained VLM model is incorporated for two main purposes: (1) to provide histopathological priors extracted from query images, and (2) to enforce morphological consistency between synthesized and ground truth images under full supervision. Furthermore, a two-stage progressive denoising strategy is introduced to enable high-resolution image synthesis.

**Strengths:**

1. The manuscript is clearly written and easy to follow. The motivations behind the proposed mechanisms for the virtual staining task are well justified and logically presented.
2. Extensive experiments on three datasets are conducted, and the reported results demonstrate strong overall performance.

**Weaknesses:**

1. The proposed method adopts one-step diffusion models for virtual staining, a specific case of pix2pix-style translation. To overcome inherent limitations, several learning mechanisms, such as introducing image priors via a pre-trained VLM, adversarial training, and 2-stage progressive training, are integrated into the baseline. While the results are promising, the overall contribution appears to be primarily an effective engineering integration of established techniques for a well-defined problem. Hence, the theoretical novelty seems limited, making this work potentially better suited for application-oriented venues like MICCAI.
2. In Figure 2, the output of the VAE is depicted as images, which is misleading since VAE latents should be numerical representations rather than image outputs. Updating Figure 2 to improve accuracy is recommended.
3. It appears that the VAE remains frozen during the entire training process. It is unclear whether the pre-trained VAE is sufficient to capture histopathological features without fine-tuning. As prior studies have shown that VAE quality strongly affects synthesis outcomes, it would be valuable to compare results with and without VAE fine-tuning.
4. Although the paper introduces a two-stage training strategy to support high-resolution synthesis for WSI, the generated images (1024 resolution) are still much smaller than real WSIs. If applied to full-scale WSIs, how many stages would be required? While the approach is promising, scalability remains an open concern.
5. The images in Figure 3 are too small to effectively demonstrate visual results, which limits their interpretability.

**Questions:**

I would like to hear feedback for the following comments:
1. Prior studies have shown that VAE quality strongly affects synthesis outcomes, why did authors still decide to freeze the VAE and only finetune the DM component? Is there any evidence to support this selection?
2. Although the paper introduces a two-stage training strategy to support high-resolution synthesis for WSI, the generated images (1024 resolution) are still much smaller than real WSIs. If applied to full-scale WSIs, how many stages would be required? While the approach is promising, scalability remains an open concern.

---

> ### Author Response · Authors · 2025-11-23
> **Response to Reviewer 6Jo9 (Part 1/2)**
>
> > **W1**: The proposed method adopts one-step diffusion models for virtual staining, a specific case of pix2pix-style translation. To overcome inherent limitations, several learning mechanisms, such as introducing image priors via a pre-trained VLM, adversarial training, and 2-stage progressive training, are integrated into the baseline. While the results are promising, the overall contribution appears to be primarily an effective engineering integration of established techniques for a well-defined problem. Hence, the theoretical novelty seems limited, making this work potentially better suited for application-oriented venues like MICCAI.
>
> **A1**: We thank the reviewer for **recognizing our significant contributions**, including our efficient and practical engineering execution, well-motivated methodology, and comprehensive, well-grounded experiments.
> Thus, we respectfully yet firmly argue that **MagicStain is not merely a technically sound reuse of existing components**, but a **concept-inspired, problem-driven advancement** that enables previously unattainable capabilities and delivers **application-level novelty** for clinicians.
> Our focus on applied technical development does not diminish the impact of our work, especially given that **it offers valuable insights aligned with the AI-for-Science emphasis highlighted by ICLR**.
>
> &emsp;
> Most notably, the simple yet effective single-step diffusion model is the key to achieving efficient, high-fidelity, and high-resolution virtual staining.
> Despite advances in efficient diffusion models for image generation, we are the first to demonstrate strong performance of a single-step diffusion model for virtual staining, as detailed in "Diffusion-based Virtual Staining" of Section 2 "RELATED WORK".
> The motivations for our domain-specific modules and loss functions are discussed in Lines 089–095: "...IHC images generated by VIMs appear hardly realistic...We attribute this to pix2pix-Turbo’s limited domain expertise in pathology...".
> Therefore, we use PathCLIP and H-channel guidance to address (1) the lack of domain knowledge and (2) the absence of explicit pathological and structural constraints, thereby overcoming the core challenge of applying a single-step diffusion model to virtual staining.
> In addition, inspired by the concept of progressive learning—from easy (low-resolution) to difficult (high-resolution) tasks—and motivated by the absence of high-resolution generation in recent single-step virtual staining models, we propose a two-stage training strategy for high-resolution adaptation (detailed in Lines 096–099).
> Our ablation study in Table 3 shows that **each component we introduce—driven by specific problems and objectives—is effective**, validating our motivation and design principles.
>
>
> > **W2**: In Figure 2, the output of the VAE is depicted as images, which is misleading since VAE latents should be numerical representations rather than image outputs. Updating Figure 2 to improve accuracy is recommended.
>
> **A2**: We thank the reviewer for the detailed, helpful suggestions.
>
> To address these points, we have revised Figure 2 as follows:
> We replace the latent-space representations (the output of the VAE encoder and the input of the VAE decoder) with color-enhanced images from the “Target Domain” and “Source Domain” (similar to Figure 3 in [1]), and we clarify in the caption that these color-enhanced images serve as schematic illustrations of latent-space representations.
>
> - [1] Sauer, Axel, Frederic Boesel, Tim Dockhorn, Andreas Blattmann, Patrick Esser, and Robin Rombach. "Fast high-resolution image synthesis with latent adversarial diffusion distillation." In SIGGRAPH Asia 2024 Conference Papers, pp. 1-11. 2024.

---

> ### Author Response · Authors · 2025-11-23
> **Response to Reviewer 6Jo9 (Part 2/2)**
>
> > **W3**: It appears that the VAE remains frozen during the entire training process. It is unclear whether the pre-trained VAE is sufficient to capture histopathological features without fine-tuning. As prior studies have shown that VAE quality strongly affects synthesis outcomes, it would be valuable to compare results with and without VAE fine-tuning.
> > **Q1**: Prior studies have shown that VAE quality strongly affects synthesis outcomes, why did authors still decide to freeze the VAE and only finetune the DM component? Is there any evidence to support this selection?
>
> **A3**: We appreciate the reviewer's insightful comments.
>
> We have investigated the influence of fine-tuning the VAE in Appendix “C.4 INFLUENCE OF FINE-TUNING VARIATIONAL AUTOENCODER (VAE).”
> As shown in Table 8, the pretrained VAE is **sufficient to capture histopathological features** for pathology image encoding and reconstruction in MagicStain, and the domain gap between natural images and pathology images has minimal impact on performance.
> Meanwhile, the impact of fine-tuning the VAE is negligible.
> We attribute this to the SD VAE’s strong encoding/decoding capacity, learned from large-scale, diverse training data that likely includes pathology images or visually similar samples.
> To **reduce training cost**, we therefore refrain from further training the VAE.
>
> > **W4, Q2**: Although the paper introduces a two-stage training strategy to support high-resolution synthesis for WSI, the generated images (1024 resolution) are still much smaller than real WSIs. If applied to full-scale WSIs, how many stages would be required? While the approach is promising, scalability remains an open concern.
>
> **A4**: We thank the reviewer for the insightful comment.
>
> We provide qualitative analyses of full-scale WSI generation results in Figures 5, 6, 7, 8, along with expert pathology evaluations in Appendix “D ADDITIONAL WSI RESULTS.”
> We use models trained solely with our two-stage training strategy, without any additional training steps, to perform inference on full-scale WSIs.
> Because WSIs are ultra-high-resolution and direct inference on the entire image is infeasible, each WSI is partitioned into non-overlapping 1024×1024 patches.
> We perform inference on each patch and then stitch the patch outputs back together to reconstruct a full WSI.
> Thus, for full-scale WSI inference, our **two-stage training strategy alone is sufficient**.
>
> According to the pathologist’s feedback, among all methods, MagicStain produced the most stable and realistic virtual staining results.
> It achieves high visual quality, with staining colors closely matching those of real HER2-IHC and H&E images, while preserving critical structure and pathology in the source images.
> Although a mild mosaic (“checkerboard”) artifact is occasionally observed—mostly in background regions, it does not affect diagnostic interpretation.
>
> > **W5**: The images in Figure 3 are too small to effectively demonstrate visual results, which limits their interpretability.
>
> **A5**: We thank the reviewer for the detailed, helpful suggestions.
>
> The original images shown in Figure 3 are 1024×1024 pixels; but due to the page-length constraint, they were shrunk to small sizes in the PDF.
> The qualitative visualizations in Figures 4 also follow the same setup.
> However, these images can be viewed in detail by zooming in for clarity.
> To improve their interpretability, we have revised the captions of Figures 3 and 4 by adding “(best viewed when zoomed in)” to explicitly guide readers to enlarge the figures for better qualitative assessment.

---

> > ### Comment · Reviewer_6Jo9 · 2025-11-26
> > **Thanks for the response.**
> >
> > The authors have addressed most of my concerns from a technical perspective, so I am willing to raise my score. However, the limited methodological innovation remains a significant concern.

---

> > > ### Author Response · Authors · 2025-12-04
> > > **Thanks for your thoughtful review.**
> > >
> > > ​We are pleased to know that our response has addressed your questions, and thank you very much for your thoughtful review.

---

### Official Review · Reviewer_WqrY · 2025-10-31

**Soundness:** 2
**Presentation:** 2
**Contribution:** 2
**Rating:** 4
**Confidence:** 3

**Summary:**

The paper introduces MagicStain, a single-step diffusion framework for virtual staining in digital pathology. Virtual staining aims to computationally translate an image of a tissue section from one stain modality (e.g., H&E) to another (e.g., IHC, Masson’s trichrome), providing a cost-effective and non-destructive alternative to chemical staining.
Traditional GAN-based models are efficient but often lack fidelity, while diffusion models produce more photorealistic results at the cost of heavy sampling (dozens of steps) and long inference times. To bridge this trade-off, MagicStain builds on a pretrained single-step diffusion backbone (e.g., SD-Turbo or similar latent diffusion variant) and adapts it for the pathology domain via three key contributions:
1. Pathology-aware conditioning using a pretrained vision–language model (PathCLIP): Introduces semantic priors reflecting cellular and histological structures, improving the alignment between input and generated stain styles.
2. Pathology- and structure-consistency losses: Applies supervision in both RGB and H-channel (hematoxylin channel) space to preserve nuclei morphology and tissue micro-architecture during stain translation.
3. Progressive high-resolution adaptation: Proposes a two-stage training pipeline (512 → 1024 px) using LoRA fine-tuning to efficiently scale up to gigapixel-level pathology slides while maintaining learned priors.

**Strengths:**

1. Practical and clinically relevant problem
- The motivation is clear and well grounded with virtual staining, which has direct diagnostic impact.
2. Strong engineering execution
- The single-step diffusion adaptation is efficient by adapting other well-pretrained models, and yields large speed-ups.
3. Integration of domain knowledge
- The use of PathCLIP priors and H-channel supervision meaningfully connects pathology semantics with generative modeling.

**Weaknesses:**

1. Incremental conceptual novelty
- The adaptation of single-step diffusion and domain priors is technically sound but conceptually incremental, as it more tends to applicational work, instead of theorectically insights.

2. Lack of theoretical justification:
- No analysis of why PathCLIP guidance and H-channel losses interact synergistically with diffusion dynamics. The approach appears largely empirical. It will be great to have an ablation that evaluate the integration of PathCLIP embeddings is helping or not.

3. Dependence on pretrained backbones:
-Performance may rely heavily on the pretrained single-step model’s capabilities (e.g., SD-Turbo). The contribution is then more about fine-tuning heuristics than architectural advancement.

**Questions:**

1. How does MagicStain differ from prior single-step or distilled diffusion models like LDM-Turbo or EfficientDM, beyond domain-specific losses?
2. What is the quantitative speed-accuracy trade-off versus few-step diffusion (e.g., 2–4 steps)? Does single-step truly match their quality, or is there a perceptual compromise?
3. How critical are the PathCLIP priors? Is there any experiments that can show the importance of it? Can the model function without them, or do they primarily anchor the translation?

---

> ### Author Response · Authors · 2025-11-23
> **Response to Reviewer WqrY (Part 1/3)**
>
> > **W1**: Incremental conceptual novelty
> > > The adaptation of single-step diffusion and domain priors is technically sound but conceptually incremental, as it more tends to applicational work, instead of theorectically insights.
>
> **A1**: We thank the reviewer for **recognizing our significant contributions**, including our efficient and practical engineering execution, well-motivated methodology, and comprehensive, well-grounded experiments.
> Thus, we respectfully yet firmly argue that **MagicStain is not merely a technically sound reuse of existing components**, but a **concept-inspired, problem-driven advancement** that enables previously unattainable capabilities and delivers **application-level novelty** for clinicians.
> Our focus on applied technical development does not diminish the impact of our work, especially given that **it offers valuable insights aligned with the AI-for-Science emphasis highlighted by ICLR**.
>
> &emsp;
> Most notably, the simple yet effective single-step diffusion model is the key to achieving efficient, high-fidelity, and high-resolution virtual staining.
> Despite advances in efficient diffusion models for image generation, we are the first to demonstrate strong performance of a single-step diffusion model for virtual staining, as detailed in "Diffusion-based Virtual Staining" of Section 2 "RELATED WORK".
> The motivations for our domain-specific modules and loss functions are discussed in Lines 089–095: "...IHC images generated by VIMs appear hardly realistic...We attribute this to pix2pix-Turbo’s limited domain expertise in pathology...".
> Therefore, we use PathCLIP and H-channel guidance to address (1) the lack of domain knowledge and (2) the absence of explicit pathological and structural constraints, thereby overcoming the core challenge of applying a single-step diffusion model to virtual staining.
> In addition, inspired by the concept of progressive learning—from easy (low-resolution) to difficult (high-resolution) tasks—and motivated by the absence of high-resolution generation in recent single-step virtual staining models, we propose a two-stage training strategy for high-resolution adaptation (detailed in Lines 096–099).
> Our ablation study in Table 3 shows that **each component we introduce—driven by specific problems and objectives—is effective**, validating our motivation and design principles.
>
> > **W2**: Lack of theoretical justification:
> > > No analysis of why PathCLIP guidance and H-channel losses interact synergistically with diffusion dynamics. The approach appears largely empirical. It will be great to have an ablation that evaluate the integration of PathCLIP embeddings is helping or not.
> > **Q.3**: How critical are the PathCLIP priors? Is there any experiments that can show the importance of it? Can the model function without them, or do they primarily anchor the translation?
>
> **A2**: Thank you for this important point.
>
> As noted in Section "INTRODUCTION", MagicStain incorporates PathCLIP guidance and H-channel losses to compensate for limited domain expertise in pathology, including pathological semantics and cellular morphology.
> Here, PathCLIP, acting as an “expert’’ assistant, encodes source images into pathology-prior embeddings that guide the I2I translation via multi-head attention.
> MagicStain further enforces pathological accuracy by using PathCLIP to extract embeddings from both generated and real target-domain images, applying a loss that encourages their similarity.
> The H-channel, which reflects nuclear morphology in H&E and IHC images, serves as an additional supervisory signal that imposes structural and semantic consistency, enhancing fidelity in both structure and pathology.
>
> &emsp;
> Although these components may appear empirical, they prove effective, as shown in Table 3 of the main text.
> We ablate the contributions of PathCLIP priors—including both the guidance for I2I translation (w/o prior) and the pathology-accuracy supervision (w/o $L_{Expert}$)—as well as the H-channel losses, including the H-channel MSE loss (w/o $L^H_{MSE}$) and LPIPS loss (w/o $L^H_{LPIPS}$), as reported in Table 3.
> Removing any of these components results in pronounced drops in the corresponding metrics, demonstrating the substantial benefits of PathCLIP guidance and H-channel losses in MagicStain.

---

> ### Author Response · Authors · 2025-11-23
> **Response to Reviewer WqrY (Part 2/3)**
>
> > **W3**: Dependence on pretrained backbones:
> > > Performance may rely heavily on the pretrained single-step model’s capabilities (e.g., SD-Turbo). The contribution is then more about fine-tuning heuristics than architectural advancement.
>
> **A3**: We thank the reviewer for the insightful comment.
>
> In Diffusion applications [1], using pretrained weights on generic natural images as initialization and then fine-tuning them is an effective and prevalent strategy for achieving better performance.
> Although there could be potential distribution gaps between pathology and natural images, several works [2, 3, 4] already adopted pretrained models (e.g., Stable Diffusion) as their initialization for histopathology image generation.
> Therefore, we likewise adapt pretrained SD-Turbo as the backbone of MagicStain, which provides superior capability for structure- and pathology-correct virtual staining compared with State-of-the-Art (SOTA) methods.
> We also evaluate the performance of training *without* pretrained weights on the H&E2IHC-HER2 dataset under the same setting as Table 1 in the main text.
> As shown in the table below, removing pretrained weights leads to worse results compared with using pretrained initialization.
> Nevertheless, it still achieves highly competitive performance compared with existing SOTA methods in Table 1 of the main text (only inferior to [5]), owing to our **architectural advancements**—including explicit incorporation of domain knowledge and high-resolution adaptation.
> **We have added these results in Appendix “C.10 IMPACT OF PRETRAINED WEIGHT.”**
>
> | Method | FID↓ | KID↓ | PSNR↑ | SSIM↑ |
> |-------|-------|-------|-------|-------|
> | w/o pretrain | 37.12 | 0.0081 | 18.28±3.708 | 0.407±0.122 |
> | w pretrain | 33.51 | 0.0033 | 18.58±3.668 | 0.411±0.125 |
>
> &emsp;
> Furthermore, as shown in the "Ablation Studies" and Table 3 in the main text, each architectural component contributes meaningfully to the overall performance, further validating our **architectural advancement**.
>
> > **Q.1**: How does MagicStain differ from prior single-step or distilled diffusion models like LDM-Turbo or EfficientDM, beyond domain-specific losses?
>
> **A4**: We appreciate the reviewer's insightful question.
>
> Both LDM-Turbo（SD-Turbo in the main text）and EfficientDM are designed to accelerate diffusion models.
> LDM-Turbo reduces the number of inference steps to a single step via adversarial training for text-to-image generation, while EfficientDM accelerates diffusion through weight quantization.
> In this work, we accelerate virtual-staining diffusion models from the perspective of reducing inference steps, using LDM-Turbo as the baseline architecture but introducing innovative solutions to address the lack of domain expertise in pathology and the poor generalization to higher resolutions when directly applying it.
> We thank the reviewer for pointing out EfficientDM, which inspires new directions for accelerating virtual-staining diffusion models; we will discuss EfficientDM in our final version and explore model weight quantization in future work.
> **We have added discussion of EfficientDM in the "Diffusion-based Virtual Staining" of Section 2 "RELATED WORK", and we will explore model weight quantization in future work.**
>
> - [1] Gaurav Parmar, Taesung Park, Srinivasa Narasimhan, and Jun-Yan Zhu. One-step image translation with text-to-image models. 2024
> - [2] Shikha Dubey, Yosep Chong, Beatrice Knudsen, and Shireen Y Elhabian. Vims: virtual immunohistochemistry multiplex staining via text-to-stain diffusion trained on uniplex stains. In International Workshop on Machine Learning in Medical Imaging, pp. 143–155. Springer, 2024.
> - [3] Man M Ho, Shikha Dubey, Yosep Chong, Beatrice Knudsen, and Tolga Tasdizen. F2fldm: Latent diffusion models with histopathology pre-trained embeddings for unpaired frozen section to FFPE translation. In IEEE/CVF Winter Conference on Applications of Computer Vision, pp. 4382–4391. IEEE, 2025.
> - [4] Srikar Yellapragada, Alexandros Graikos, Zilinghan Li, Kostas Triaridis, Varun Belagali, Saarthak Kapse, Tarak Nath Nandi, Ravi K Madduri, Prateek Prasanna, Tahsin Kurc, et al. Pixcell: A generative foundation model for digital histopathology images. arXiv preprint arXiv:2506.05127, 2025.
> - [5] Peng, Qiong, Weiping Lin, Yihuang Hu, Ailisi Bao, Chenyu Lian, Weiwei Wei, Meng Yue, Jingxin Liu, Lequan Yu, and Liansheng Wang. "Advancing H&E-to-IHC virtual staining with task-specific domain knowledge for HER2 scoring." In International Conference on Medical Image Computing and Computer-Assisted Intervention, pp. 3-13. Cham: Springer Nature Switzerland, 2024.

---

> ### Author Response · Authors · 2025-11-23
> **Response to Reviewer WqrY (Part 3/3)**
>
> > **Q.2**: What is the quantitative speed-accuracy trade-off versus few-step diffusion (e.g., 2–4 steps)? Does single-step truly match their quality, or is there a perceptual compromise?
>
> **A5**: We appreciate the reviewer's valuable comment.
>
> Few-step diffusion models are typically obtained by distilling time-consuming, many-step teacher models using various approaches, including consistency-based methods, adversarial learning, and variational score distillation.
> In domain-specific applications of diffusion (e.g., portrait video generation [6], image restoration [7], image super-resolution [8]), researchers often design task-specific modules or losses to facilitate few-step distillation, since directly applying these distillation approaches often leads to poor results.
> Virtual staining, as a domain-specific application, could in principle also adopt such distillation methods to obtain few-step diffusion models.
> However, to the best of our knowledge, no distillation method has yet been designed specifically for virtual staining, which limits our ability to compare model quality with such approaches.
> Regarding speed, under the same architecture and parameter budget, our single-step virtual-staining diffusion model will be faster than few-step models (e.g., 2–4 steps) due to fewer sampling iterations.
> We appreciate the reviewer’s suggestion and will explore few-step virtual-staining diffusion in future work.
>
> - [6] Guo, Hanzhong, Hongwei Yi, Daquan Zhou, Alexander William Bergman, Michael Lingelbach, and Yizhou Yu. "Real-time One-Step Diffusion-based Expressive Portrait Videos Generation." arXiv preprint arXiv:2412.13479 (2024).
> - [7] Li, Senmao, Kai Wang, Joost van de Weijer, Fahad Shahbaz Khan, Chun-Le Guo, Shiqi Yang, Yaxing Wang, Jian Yang, and Ming-Ming Cheng. "InterLCM: Low-Quality Images as Intermediate States of Latent Consistency Models for Effective Blind Face Restoration." ICLR, 2025.
> - [8] Dong, Linwei, et al. "Tsd-sr: One-step diffusion with target score distillation for real-world image super-resolution." Proceedings of the Computer Vision and Pattern Recognition Conference. 2025.

---

### Official Review · Reviewer_6V7B · 2025-11-03

**Soundness:** 2
**Presentation:** 3
**Contribution:** 2
**Rating:** 4
**Confidence:** 3

**Summary:**

MagicStain Propose single step diffusion framework adapted from prior art for H&E to IHC image-to-image translation (virtual staining). It adapts SD-Turbo and pix2pix-Turbo to injects PathCLIP pathology priors, adds H-channel (LPIPS and MSE)losses, and uses a two-stage 512 to 1024 progressive training with LoRA for high-res outputs. Experiments on three datasets report top FID/KID (and competitive PSNR/SSIM), external-set generalization, and a downstream win in classification.

**Strengths:**

1. Speed: As compared to previous methods especially the ones with multiple steps it is much faster and therefore practical to use for clinicians.
2. Well motivated and supported claims: PathCLIP prior and losses on H-channel and expert embeddings target pathology semantics are well motivated. Ablations show each piece helps giving justification for the choices made.
3. Two stage recipe for high resolution: Clear two-stage LoRA adaptation to 1024 with supporting ablations is very helpful. Working on WSI images is more clinically relevant since it follows how pathologists work with WSI (high magnification to low magnification).
4. Strong overall, external validation and downstream task: Overall the evaluation shows better results for the proposed work with downstream classification task where Magic stain wins even as compared to the real IHC images.

**Weaknesses:**

1. Metric choice not fully pathology-standard. The paper leans on FID/KID (with natural Inception?, please clarify this) and PSNR/SSIM; these do not directly quantify clinical correctness or region-level fidelity in histopathology. Authors acknowledge structural misalignment for SSIM/PSNR but still rely on them. Consider pathology encoders / clinical labelers (e.g., PLIP similarity, cell/lesion segmentation or classification).
2. “Beats Real IHC” on downstream classifier needs careful interpretation and more explanation. Surpassing the classifier trained on real IHC likely reflects cross-cohort domain shift (train on HER2, test on Ext) rather than virtual images exceeding real-image information. Mentioning single sentence about it is not enough. Please analyze shifts and add CI/significance. Also, this should be paired with performance on indomain instead of out of cohort if domain shift is not properly analyzed.
3. The paper’s novelty largely lies in applying established components to pathology. While the system is well-validated, the methodological contribution appears incremental.
4. Some claims need more evidence: For example on line 321, "Notably, lower FID and KID values suggest better pathology matches, e.g., cancerous status" need more evidence and justification. Lacking domain expert evaluation for clinical accuracy and confidence in the results presented.

**Questions:**

Q.1 Please adapt the current evaluation for Pathology and domain expert evaluation with strengthen the results provided.
Q.2. I suggest downweighing/restating some of the claims pointed out in the weakness or providing more evidence.

---

> ### Author Response · Authors · 2025-11-23
> **Response to Reviewer 6V7B (Part 1/3)**
>
> > **W1**:Metric choice not fully pathology-standard. The paper leans on FID/KID (with natural Inception?, please clarify this) and PSNR/SSIM; these do not directly quantify clinical correctness or region-level fidelity in histopathology. Authors acknowledge structural misalignment for SSIM/PSNR but still rely on them. Consider pathology encoders / clinical labelers (e.g., PLIP similarity, cell/lesion segmentation or classification).
>
> **A1**: We appreciate the reviewer's insightful comments.
>
> We compute FID/KID using the Inception Network pretrained on natural images.
> To the best of our knowledge, numerous prior works on virtual staining [1, 2, 3, 4] also computed FID/KID using Inception Network pretrained on natural-image datasets (e.g., ImageNet).
> Inception Networks trained on large-scale datasets exhibit strong generalization, enabling effective feature extraction and reliable estimation of the **disparity between the distributions** of two image groups, even in pathology scenarios.
> To obtain a more *pathology-standard* evaluation, we follow the reviewer's suggestion to additionally employ a pathology-specific encoder—PLIP—to assess the gap between the generated results and the target domains by computing the mean cosine similarity of the feature embeddings extracted by PLIP for each pair on the H&E2IHC-HER2 dataset [1].
>
> | Metric | Pix2pix | Pix2pixHD | [1] | CycleGAN | AI-FFPE | ControlNet | StainFuser | pix2pix-Turbo | MagicStain (ours) |
> |-------|-------|-------|-------|-------|-------|-------|-------|-------|-------|
> | PLIP | 0.9711 | 0.9649 | 0.9724 | 0.9641 | 0.9574 | 0.8367 | 0.9357 | 0.9572 | **0.9733** |
>
> As shown, MagicStain achieves the best PLIP similarity scores, demonstrating its superior capability in **clinically correct** virtual staining and validating its effective design and strong performance under pathology-standard evaluation.
> **We have added these results in Appendix “C.12 ADDITIONAL EVALUATION METRICS.”**
>
> &emsp;
> In addition, we conducted a clinically relevant downstream classification task to quantitatively evaluate clinical correctness in histopathology (cf. **A2**).
>
> - [1] Peng, Qiong, Weiping Lin, Yihuang Hu, Ailisi Bao, Chenyu Lian, Weiwei Wei, Meng Yue, Jingxin Liu, Lequan Yu, and Liansheng Wang. "Advancing H&E-to-IHC virtual staining with task-specific domain knowledge for HER2 scoring." In International Conference on Medical Image Computing and Computer-Assisted Intervention, pp. 3-13. Cham: Springer Nature Switzerland, 2024.
> - [2] Li, Fangda, Zhiqiang Hu, Wen Chen, and Avinash Kak. "Adaptive supervised patchnce loss for learning h&e-to-ihc stain translation with inconsistent groundtruth image pairs." In International Conference on Medical Image Computing and Computer-Assisted Intervention, pp. 632-641. Cham: Springer Nature Switzerland, 2023.
> - [3] Pushpak Pati, Sofia Karkampouna, Francesco Bonollo, Eva Comperat, Martina Radi ´ c, Martin ´ Spahn, Adriano Martinelli, Martin Wartenberg, Marianna Kruithof-de Julio, and Marianna Rapsomaniki. Accelerating histopathology workflows with generative ai-based virtually multiplexed tumour profiling. Nature Machine Intelligence, 6(9):1077–1093, 2024.
> - [4] Wang, Tong, et al. "ODA-GAN: Orthogonal Decoupling Alignment GAN Assisted by Weakly-supervised Learning for Virtual Immunohistochemistry Staining." Proceedings of the Computer Vision and Pattern Recognition Conference. 2025.

---

> > ### Comment · Reviewer_6V7B · 2025-11-27
> >
> > Thank you for running the additional experiments and for clarifying how FID/KID are computed. However, I remain unconvinced about the use of FID/KID based on an Inception network trained on natural images. While you correctly point out that several prior virtual staining works follow this convention, there are also recent studies that explicitly adopt in-domain encoders for FID/KID in histopathology [1, Sec. 6]. As a community, we should be willing to revise established practice when it does not align well with the target domain, and in this case I believe it is important to compute FID/KID using a pathology-specific encoder rather than relying solely on ImageNet Inception. I also appreciate the new PLIP-based evaluation. That said, the reported improvement over [2] is quite small (on the order of 0.0009), which may well lie within the variance of the metric. Without confidence intervals or statistical tests, it is difficult to interpret this as a robust or clinically meaningful gain, so I do not find this result alone fully compelling as evidence of a clear advantage.
> >
> >
> >
> >
> > Ref:
> > [1] https://openaccess.thecvf.com/content/ICCV2025/supplemental/Bhosale_PathDiff_Histopathology_Image_ICCV_2025_supplemental.pdf
> > [2] Peng, Qiong, Weiping Lin, Yihuang Hu, Ailisi Bao, Chenyu Lian, Weiwei Wei, Meng Yue, Jingxin Liu, Lequan Yu, and Liansheng Wang. "Advancing H&E-to-IHC virtual staining with task-specific domain knowledge for HER2 scoring." In International Conference on Medical Image Computing and Computer-Assisted Intervention, pp. 3-13. Cham: Springer Nature Switzerland, 2024.

---

> > > ### Comment · Reviewer_JMPk · 2025-11-27
> > > **Side Comments related to this Discussion On FID/KID on Domain Specific Encoders**
> > >
> > > A recent arXiv study [1] showed that FID and KID values computed using encoders trained on natural images or histopathology images are highly correlated, with a Pearson correlation coefficient greater than 0.80. This suggests that even if FID/KID are calculated using pathology-specific encoders, the metrics may not necessarily capture domain-specific pathological information. However, as the study is not yet peer-reviewed, the relevance of these results remains uncertain. [1] also showed that FID and KID are not relaible metrics for measuring staining accuracy.
> > >
> > > [1] Kataria, Tushar, Shikha Dubey, Mary Bronner, Jolanta Jedrzkiewicz, Ben J. Brintz, Shireen Y. Elhabian, and Beatrice S. Knudsen. "Building Trust in Virtual Immunohistochemistry: Automated Assessment of Image Quality." arXiv preprint arXiv:2511.04615 (2025).

---

> > > > ### Comment · Reviewer_6V7B · 2025-11-27
> > > > **Thanks to the reviewer JMPk**
> > > >
> > > > Thank you for pointing this out, indeed the overall evaluation protocol should be improved. I will consider this while giving the final rating.

---

> ### Author Response · Authors · 2025-11-23
> **Response to Reviewer 6V7B (Part 2/3)**
>
> > **W2**:"Beats Real IHC" on downstream classifier needs careful interpretation and more explanation. Surpassing the classifier trained on real IHC likely reflects cross-cohort domain shift (train on HER2, test on Ext) rather than virtual images exceeding real-image information. Mentioning single sentence about it is not enough. Please analyze shifts and add CI/significance. Also, this should be paired with performance on indomain instead of out of cohort if domain shift is not properly analyzed.
>
> **A2**: We thank the reviewer for the insightful comment.
>
> The IHC images in H&E2IHC-HER2 are processed using the **4B5** antibody, while the IHC images in H&E2IHC-Ext are stained with the **SP3** or **CB11** antibodies.
> As we train the classifiers on 4B5 IHC images (either real or virtually stained), these images are considered **in-domain**, whereas the SP3 and CB11 IHC images constitute **cross-domain** data.
> To evaluate the in-domain performance, we now additionally collect 285 4B5 IHC images (the same number of images as in the H&E2IHC-Ext dataset) with no overlap with the H&E2IHC-HER2 data, forming an **independent in-domain test set**.
> We then evaluate the in-domain performance by applying  the 4B5-trained classifiers to the in-domain test set.
>
> | Model | ACC↑ | F1↑ |
> |-------|-------|-------|
> | Real IHC | 0.8456 | 0.8160 |
> | MagicStain | 0.8421 | 0.8171 |
>
> As shown above, the performance is overall better than that on the cross-domain data, as expected.
> In addition, the classifier trained on MagicStain-generated 4B5 IHC images achieves performance **competitive with** that of the classifier trained on real 4B5 IHC images.
> This indicates that the IHC images generated by MagicStain exhibit **accurate HER2 expression**, i.e., accurate pathological translation from H&E to IHC images.
> **We have added these results in Appendix “C.11 DOWNSTREAM TASK PERFORMANCE ON IN-DOMAIN DATA.”**
>
> > **W3**: The paper’s novelty largely lies in applying established components to pathology. While the system is well-validated, the methodological contribution appears incremental.
>
> **A3**: We thank the reviewer for **recognizing our significant contributions**, including our efficient and practical engineering execution, well-motivated methodology, and comprehensive, well-grounded experiments.
> Thus, we respectfully yet firmly argue that **MagicStain is not merely a technically sound reuse of existing components**, but a **concept-inspired, problem-driven advancement** that enables previously unattainable capabilities and delivers **application-level novelty** for clinicians.
> Our focus on applied technical development does not diminish the impact of our work, especially given that **it offers valuable insights aligned with the AI-for-Science emphasis highlighted by ICLR**.
>
> &emsp;
> Most notably, the simple yet effective single-step diffusion model is the key to achieving efficient, high-fidelity, and high-resolution virtual staining.
> Despite advances in efficient diffusion models for image generation, we are the first to demonstrate strong performance of a single-step diffusion model for virtual staining, as detailed in "Diffusion-based Virtual Staining" of Section 2 "RELATED WORK".
> The motivations for our domain-specific modules and loss functions are discussed in Lines 089–095: "...IHC images generated by VIMs appear hardly realistic...We attribute this to pix2pix-Turbo’s limited domain expertise in pathology...".
> Therefore, we use PathCLIP and H-channel guidance to address (1) the lack of domain knowledge and (2) the absence of explicit pathological and structural constraints, thereby overcoming the core challenge of applying a single-step diffusion model to virtual staining.
> In addition, inspired by the concept of progressive learning—from easy (low-resolution) to difficult (high-resolution) tasks—and motivated by the absence of high-resolution generation in recent single-step virtual staining models, we propose a two-stage training strategy for high-resolution adaptation (detailed in Lines 096–099).
> Our ablation study in Table 3 shows that **each component we introduce—driven by specific problems and objectives—is effective**, validating our motivation and design principles.

---

> > ### Comment · Reviewer_6V7B · 2025-11-27
> >
> > Thank you for providing the additional classification results. I recommend clarifying these findings in the main paper as well and explicitly directing the reader to the supplementary section for details. I would also suggest softening the claim that the method “beats real IHC,” since the reported accuracy does not clearly exceed that of a classifier trained on real data. Moreover, the observed gains are very small in magnitude and could plausibly fall within the variance of the metric; without statistical significance tests and a more thorough evaluation, this claim should be down-weighted.

---

> > > ### Comment · Reviewer_6V7B · 2025-11-27
> > >
> > > Overall, the authors have partially addressed my concerns. However, I remain unconvinced about the methodological novelty of the work beyond its specific instantiation on a histopathology task (PLIP prior, H Loss and their verification in ablations are still incremental contributions). Most notably, I am not fully persuaded by the reported performance gains over prior methods, particularly given the remaining weaknesses in the evaluation protocol. That said, I appreciate the authors’ efforts to address several of the raised points and to include domain-expert assessments, and I am willing to consider a modest increase in my rating in light of these additions.

---

> ### Author Response · Authors · 2025-11-23
> **Response to Reviewer 6V7B (Part 3/3)**
>
> > **W4**: Some claims need more evidence: For example on line 321, "Notably, lower FID and KID values suggest better pathology matches, e.g., cancerous status" need more evidence and justification. Lacking domain expert evaluation for clinical accuracy and confidence in the results presented.
> > **Q.1**: Please adapt the current evaluation for Pathology and domain expert evaluation with strengthen the results provided.
> > **Q.2**. I suggest downweighing/restating some of the claims pointed out in the weakness or providing more evidence.
>
> **A4**: Thank you for this important point.
> As discussed in **A1**, the perception-oriented metrics (FID and KID) have been widely adopted in numerous prior works on virtual staining [1, 2, 3, 4].
> As noted in these studies, lower perception-oriented metric scores indicate a smaller disparity between the distributions of generated results and the target domains, thereby reflecting the model’s ability to produce **pathological feature distributions** that are closely aligned with real data.
> Prior works have demonstrated that their proposed methods outperform State-of-the-Art (SOTA) approaches in the pathological accuracy of generated results (e.g., [1] validated that their generated results contain accurate pathological information, such as HER2 expression, via downstream tasks) and achieve the best perception-oriented metric scores reported in their papers.
> Therefore, prior studies have already established that perception-oriented metrics can serve as effective indicators of pathological accuracy.
>
> &emsp;
> To further validate the degree of pathology matching in MagicStain’s generated results, we additionally provide expert pathological assessment in the APPENDIX "D ADDITIONAL WSI RESULTS".
> We selected the compared methods, along with MagicStain, to generate IHC or H&E WSIs from the corresponding H&E or FFPE WSIs.
> The generated results were then **provided to a board-certified pathologist for domain-specific evaluation**, including an overall, blind ranking on a scale of 1 (best) to 4 (worst).
> The mean ranking is: MagicStain (1.0), pix2pix-Turbo (2.0), Pix2pixHD (3.0), and AI-FFPE (4.0).
> According to the pathologist’s feedback, MagicStain produces the most stable and realistic virtual staining results among all methods and preserves critical structures and pathological features from the source images. In contrast, the compared methods exhibit suboptimal staining quality.
> These conclusions regarding pathological accuracy on WSIs are consistent with the perception-oriented metric scores of MagicStain and the baseline methods.
>
> &emsp;
> In summary, supported by the lowest perception-oriented metrics (FID and KID) scores, the highest PLIP similarity in **A1**, and domain expert evaluation, we conclude that **MagicStain produces results most closely aligned with the target domains in terms of pathological feature distributions**.
>
> - [1] Peng, Qiong, Weiping Lin, Yihuang Hu, Ailisi Bao, Chenyu Lian, Weiwei Wei, Meng Yue, Jingxin Liu, Lequan Yu, and Liansheng Wang. "Advancing H&E-to-IHC virtual staining with task-specific domain knowledge for HER2 scoring." In International Conference on Medical Image Computing and Computer-Assisted Intervention, pp. 3-13. Cham: Springer Nature Switzerland, 2024.
> - [2] Li, Fangda, Zhiqiang Hu, Wen Chen, and Avinash Kak. "Adaptive supervised patchnce loss for learning h&e-to-ihc stain translation with inconsistent groundtruth image pairs." In International Conference on Medical Image Computing and Computer-Assisted Intervention, pp. 632-641. Cham: Springer Nature Switzerland, 2023.
> - [3] Pushpak Pati, Sofia Karkampouna, Francesco Bonollo, Eva Comperat, Martina Radi ´ c, Martin ´ Spahn, Adriano Martinelli, Martin Wartenberg, Marianna Kruithof-de Julio, and Marianna Rapsomaniki. Accelerating histopathology workflows with generative ai-based virtually multiplexed tumour profiling. Nature Machine Intelligence, 6(9):1077–1093, 2024.
> - [4] Wang, Tong, et al. "ODA-GAN: Orthogonal Decoupling Alignment GAN Assisted by Weakly-supervised Learning for Virtual Immunohistochemistry Staining." Proceedings of the Computer Vision and Pattern Recognition Conference. 2025.

---

> ### Author Response · Authors · 2025-12-03
> **Response to Reviewer 6V7B**
>
> Thank you for your appreciation of our detailed rebuttal.
>
> > **Q3**: It is important to compute FID/KID using a pathology-specific encoder rather than relying solely on ImageNet Inception.
>
> **A5**: We appreciate the reviewer's excellent suggestion.
> Following [1], we compute FID using the pathology-specific encoder UNI2-h.
>
> | Metric | Pix2pix | Pix2pixHD | [2] | CycleGAN | AI-FFPE | ControlNet | StainFuser | pix2pix-Turbo | MagicStain (ours) |
> |-------|-------|-------|-------|-------|-------|-------|-------|-------|-------|
> | FID-UNI2-h↓ | 107.12 | 109.52 | 90.883 | 90.593 | 87.001 | 323.28 | 145.21 | 186.39 | **77.764** |
>
> &emsp;
> As shown above, MagicStain achieves consistently superior performance across various SOTA methods, demonstrating the advantage of our approach.
> We have added these results in Appendix “C.12 ADDITIONAL EVALUATION METRICS.”
>
> > **Q4**: The reported improvement over [2] is quite small (on the order of 0.0009), which may well lie within the variance of the metric. Without confidence intervals or statistical tests, it is difficult to interpret this as a robust or clinically meaningful gain, so I do not find this result alone fully compelling as evidence of a clear advantage.
>
> **A6**: We appreciate the reviewer's excellent suggestion.
>
> We conduct the Wilcoxon signed-rank test for pairwise comparison with our method on PLIP scores, using “*” to denote $p < 0.05$ in the table.
>
> | Metric | Pix2pix | Pix2pixHD | [2] | CycleGAN | AI-FFPE | ControlNet | StainFuser | pix2pix-Turbo | MagicStain (ours) |
> |-------|-------|-------|-------|-------|-------|-------|-------|-------|-------|
> | PLIP↑ | 0.9711* | 0.9649* | 0.9724 | 0.9641* | 0.9574* | 0.8367* | 0.9357* | 0.9572* | 0.9733 |
>
> &emsp;
> As shown, our method attains statistically significant improvements ($p < 0.05$) over most existing SOTA methods, indicating that the higher PLIP scores achieved by MagicStain are reliable.
> Although MagicStain and [2] obtain comparable PLIP scores, MagicStain exhibits superior performance in both downstream tasks and other metrics of virtual staining tasks compared with [2], as shown in Tables 1 and 5, further substantiating the overall advantage of MagicStain.
> We have added these results in Appendix “C.12 ADDITIONAL EVALUATION METRICS.”
>
> > **Q5**: Suggestion about clarifying the findings of the additional classification results in the main paper as well and explicitly directing the reader to the supplementary section for details.
>
> **A7**: We appreciate the reviewer's excellent suggestion.
> We have added clarification and explicitly reference to “Evaluation on Downstream Task” in Section “4 EXPERIMENTS”.
>
> > **Q6**: Suggestion about softening the claim that the method “beats real IHC,” since the reported accuracy does not clearly exceed that of a classifier trained on real data.
>
> **A8**: We appreciate the reviewer's excellent suggestion.
> We have revised the statement to: “The classifier trained on IHC images generated by MagicStain achieves performance comparable to that obtained using real IHC images.”
>
> - [1] https://openaccess.thecvf.com/content/ICCV2025/supplemental/Bhosale_PathDiff_Histopathology_Image_ICCV_2025_supplemental.pdf
> - [2] Peng, Qiong, Weiping Lin, Yihuang Hu, Ailisi Bao, Chenyu Lian, Weiwei Wei, Meng Yue, Jingxin Liu, Lequan Yu, and Liansheng Wang. "Advancing H&E-to-IHC virtual staining with task-specific domain knowledge for HER2 scoring." In International Conference on Medical Image Computing and Computer-Assisted Intervention, pp. 3-13. Cham: Springer Nature Switzerland, 2024.

---

### Author Response · Authors · 2025-12-03
**Review and Reviewer-Author Discussion Summary (Part 2/2)**

- **Concerns about experiments.**
- - Downstream task on indomain. (6V7B W2)

    **Our Response.**
    We evaluate the in-domain performance, showing that the IHC images generated by MagicStain achieves competitive results with real IHC images.

  - Qualitative visualizations in downstream tasks. (JMPk Q5)

    **Our Response.**
    We provide qualitative examples and further show that HER2 scores 1+ and 2+ are particularly challenging for existing SOTA methods, whereas MagicStain successfully distinguishes between them.

- **Regarding concerns stemming from misunderstandings of the manuscript.**
- - Applying progressive training here is not logically sound. (JMPk W3)
  - Choice of Wilcoxon signed-rank or simple t-test. (JMPk W4)
  - The process and interpretation of downstream task results. (JMPk W5, Q5)
  - Manual evaluation of IHC WSI images. (JMPk W6)
  - Comparing results with and without VAE fine-tuning. (6Jo9 W3)
  - The applicability of MagicStain to full-scale WSIs. (6Jo9 W4, Q2)
  - The theoretical justification and ablation for PathCLIP and H-channel. (WqrY W2, Q3)
  - Difference between MagicStain and prior single-step or distilled diffusion models like LDM-Turbo or EfficientDM. (WqrY Q1)
  - Adapting the Pathology and domain expert evaluation. (6V7B Q1)

  We respectfully yet firmly clarify that these points have already been described or mentioned in the main text or the appendix of the manuscript submitted in our first, initial submission.
  Accordingly, in the rebuttal, we provided reviewers with a more detailed clarification of these points.

- **Presentation issues.**
- - The output of the VAE is depicted as images in Figure 2. (6Jo9 W2)
  - The images in Figure 3 are too small. (6Jo9 W5)
  - Softening the claim that lower FID and KID values suggest better pathology matches. (6V7B W4, Q2)
  - Adding explicit reference to the supplementary section for details of additional classification results. (6V7B Q5)
  - Softening the claim that the method “beats real IHC,”. (6V7B Q6)

  We thank the reviewers for their excellent suggestions for improving our paper, and we have already incorporated the corresponding revisions.

Overall, we have thoroughly addressed all questions and comments raised by the reviewers and have updated the manuscript accordingly, following their suggestions, including fixes to presentation issues and the addition of new experiments.

Again, thank you for the time and thoughtful reviews you devoted to this paper. Your efforts have made this paper even stronger.

Authors

---

### Author Response · Authors · 2025-12-03
**Review and Reviewer-Author Discussion Summary (Part 1/2)**

Dear PCs, SACs, ACs, and Reviewers,

Thank you very much for your valuable contributions to our work.
To assist the newly assigned AC and help reduce their workload, we provide below a summary of the key points from the reviews and the reviewer-author discussions.

**Strength:** Overall, we are grateful that the reviewers gave this paper a highly positive evaluation. Specifically:

- **Efficient, effective, and practical engineering design** tailored for clinical use, especially for workflows involving WSI images.
- A **well-motivated and impactful methodology**, with clearly justified mechanisms and a logically presented narrative, making the manuscript easy to follow.
- **Comprehensive and well-grounded experiments** across three datasets, including strong overall results, external validation, downstream tasks, and extensive ablations.

**Concerns and Our Addressing:** During the discussion period, we actively addressed the reviewers' concerns.
We are pleased that on Nov. 27, 2025, reviewers 6V7B, 6Jo9, and JMPk have stated that our rebuttal resolved the majority of their concerns.
We thank 6V7B and JMPk for subsequently providing additional valuable comments.
However, due to ICLR’s policy, reviewers cannot respond to authors thereafter.
Here, we summarize all issues alongside our corresponding solutions and outcomes.
Please refer to our responses to the respective reviewers for details.

- **Concerns about theoretical novelty.**

We respectfully yet firmly argue that **MagicStain is not merely a technically sound reuse of existing components**, but a **concept-inspired, problem-driven advancement** that enables previously unattainable capabilities and delivers **application-level novelty** for clinicians.
Our focus on applied technical development does not diminish the impact of our work, especially given that **it offers valuable insights aligned with the AI-for-Science emphasis highlighted by ICLR**.

- **Concerns about experiments.**
- - Metric choice. (6V7B W1, Q3; JMPk Q4)

    **Our Response.**
    We first clarify that using FID and KID is also reasonable, as many prior works have adopted these metrics.
    We then incorporate additional evaluations—including DICE, IoU, TPR, TNR, FID based on UNI2-h, and the PLIP score—to validate the virtual staining task, including comparisons with SOTA methods or ablations of key components.
    Across all experiments, results consistently demonstrate the advantages of MagicStain and the contributions of our proposed components.

  - Whether vision-only encoders outperform PathCLIP? (JMPk W1)

    **Our Response.**
    We replace PathCLIP with UNI2-h or Virchow2 and evaluate their performance on the virtual staining task.
    Results show that our chosen VLM-based PathCLIP achieves superior performance and confirm that MagicStain remains compatible with other histopathology-pretrained models.

  - Emphasizing the H-channel might produce inaccurate results for the DAB channel. (JMPk W2, Q4)

    **Our Response.**
    We use DICE, IoU, TPR, and TNR to assess the impact of H-channel losses on DAB channel accuracy.
    Results show that enhancing the H channel does not compromise the accuracy of the DAB-channel.

  - Discrepancy of training epochs between MagicStain (20 epochs) and other models (200 epochs). (JMPk Q1)

    **Our Response.**
    We train MagicStain for 200 epochs and obtain results comparable to those achieved with only 20 epochs.
    This shows that MagicStain converges within just 15–20 epochs while still delivering strong performance, thanks to our architectural advancements.

  - Comparisons with more existing methods. (JMPk Q2)

    **Our Response.**
    We compare MagicStain with ASP and Pyramidpix2pix, and the results show that MagicStain achieves superior performance.

  - Results on additional public datasets for reproducibility. (JMPk Q3)

    **Our Response.**
    We conducted additional experiments on widely used public datasets, including BCI and the four MIST datasets (HER2, ER, Ki67, PR).
    The results again demonstrate that MagicStain consistently outperforms competing SOTA methods.

  - Dependence on pretrained backbones. (WqrY W3)

    **Our Response.**
    We explained the reason for using pretrained weights and further evaluated performance when training without pretrained weights.
    The results indicate that MagicStain has low dependency on pretrained backbones, and its strong performance primarily stems from our architectural advancement.

  - Quantitative speed-accuracy trade-off versus few-step diffusion. (WqrY Q2)

    **Our Response.**
    We clarify that single-step models will inevitably be faster than few-step diffusion models under the same architecture and parameter budget.
    The current quality is difficult to compare due to the lack of few-step virtual staining diffusion models.

---

### Meta-Review · Area_Chair_FJ9T · 2026-01-02

**Summary:**

Overall, reviewers agree that the paper presents a well-engineered, clearly written, and practically motivated system for virtual staining, with strengths in computational efficiency (single-step diffusion), strong empirical results across multiple datasets, extensive ablations, and clinically relevant evaluations, including downstream tasks and expert assessment. The integration of domain priors, H-channel constraints, and progressive high-resolution training was generally viewed as effective and carefully designed.

Howver, all the reviewers raised concerns about limited novelty, regarding this work as an incremental integration of existing techniques (single-step diffusion, pretrained backbones, domain-specific losses) rather than a fundamentally new methodological contribution. Additional weaknesses include insufficient theoretical justification, heavy reliance on pretrained models, evaluation concerns (metric suitability, small performance margins, lack of strong statistical evidence), and reproducibility issues. Some of them have been addressed in the rebuttal. However, the datasets used in this work are not publicly available and the authors cannot release them. Therefore, many experiments may not be reproducible.

**Reviewer Concerns:**

>**What the rebuttal addressed:**

The rebuttal addressed many of the technical and experimental concerns raised by the reviewers. The authors added a number of missing experiments and ablations, clarified training choices, included additional baselines, and expanded the evaluation with pathology-specific metrics and downstream tasks. They also softened some over-strong claims (such as outperforming real IHC), added statistical tests, and provided expert qualitative assessments on WSIs. Overall, these additions helped clarify how the system behaves in practice, and the questions about evaluation protocol and implementation details were largely resolved.

>**What remains unresolved:**

The rebuttal does not really change the bigger-picture concerns. The main issue of limited novelty remains: the method is still largely seen as an effective combination of existing ideas, i.e., single-step diffusion, pretrained backbones, and domain-specific losses, rather than a clear conceptual or theoretical improvement. It is not very convincing that the reported performance gains are strong or robust, as the improvements are often small and sensitive to evaluation choices. In addition, while the authors provide empirical ablations, the underlying design choices (e.g., why PathCLIP guidance and H-channel losses should work together in a principled way) are still not well justified. Finally, concerns around generalization, reliance on pretrained models, and reproducibility have not been fully addressed especially datasets will not be released.

**Reviewer Scores:**

Initially, all the reviewers gave negative ratings due the limited technical novelty. After reading the authors' rebuttal, some of the concerns espeically on the details and metrics have been clarified. However, the rebuttal of the technical novelty does not convince AC as the performance improvement is considered as marginal, and the datasets used for ablation study cannot be released.

---

### Decision · Program_Chairs · 2026-01-26

Reject